



# Evaluating porosity estimates for sandstones based on X-ray micro-tomographic images

Mathias Nehler[1], Ferdinand Stoeckhert[2], Anne Oelker[3], Jörg Renner[2], and Erik Saenger[1,2]

[1]International Geothermal Centre, Bochum University of Applied Sciences, 44801, Germany.
[2]Ruhr-Universität, Bochum, 44801, Germany.
[3]RWTH Aachen, Aachen, 52062, Germany

*Correspondence to:* Mathias Nehler (mathias.nehler@hs-bochum.de)

**Abstract.** We compare experimentally determined porosity with values derived from X-ray tomography for a suite of eight sandstone varieties covering a porosity range from about 3 to 25 %. In addition, we performed conventional stereological analysis of SEM images and examined thin sections. We investigated the sensitivity of segmentation, the conversion of the tomographic gray-value images representing the obtained X-ray attenuation coefficients into binary images, to (a) resolution of the digital images, (b) denoising filters, and (c) seven thresholding methods. Images of sandstones with porosities of 15 to 25 % exhibit a bimodal intensity distribution of the attenuation coefficients, enabling unambiguous segmentation that gives porosity values closely matching the laboratory values. For samples with lower porosities, pores and grains do not separate well in the skewed unimodal intensity histograms. For these samples, all tested thresholding methods tend to miscalculate porosity significantly. In addition to absolute porosity, the ratio between pore size and resolution, and mineralogical composition of the rocks affect the biases of the global segmentation methods.

## 1 Introduction

Fluid transport and storage properties of rocks are related to their microstructure (e.g. Bear, 1988; Dullien, 2012; Head and Vanorio, 2016). Porosity, the ratio between void volume and total volume, constitutes one of the most important microstructural parameters owing to its relevance for fluid transport and associated processes but also for elastic and inelastic deformation behavior (e.g. Buryakovsky et al., 2005). Non-destructive structural characterization plays a crucial role in the modeling of effective rock properties, such as permeability (Shah et al., 2015), elastic moduli (Madonna et al., 2012), and electric conductivity (Andrä et al., 2013), relevant for numerous geological applications, such as two-phase flow in hydrocarbon reservoirs (Berg et al., 2013; Singh et al., 2016), freshwater provision (Taina et al., 2008; Ott et al., 2012) or geothermal energy recovery (Gualda and Rivers, 2006; Pamukcu et al., 2012). Specifically, digital rock physics relies on performing numerical experiments on digitized microstructures of real rocks (Dvorkin et al., 2011). Before it can be applied with confidence any proposed procedure





for providing such digital structures has to reproduce reliably results of laboratory measurements for the bulk microstructural parameter porosity.

Porosity can be quantified by various laboratory techniques at different scales (e.g. Anovitz and Cole, 2015). Common methods make use of volume measurements by saturation (Manger, 1963), buoyancy (Wong et al., 1999), or gas expansion (Washburn and Bunting, 1922). Additional parameters, such as pore-throat size distributions and specific surface area, are derived from mercury intrusion porosimetry (Webb, 2001) and gas adsorption (Naderi, op. 2015), respectively. A variety of two-dimensional imaging methods are also employed in quantitative stereological analyses, such as optical-light analysis of thin sections (Ehrlich and Kennedy, 1984), scanning electron microscopy (SEM, Camp et al., 2013) or transmission electron microscopy (TEM). Tomographic imaging methods using X-rays, neutrons or electrons passing through a material allow for a three-dimensional differentiation of phases in a material based on their attenuation characteristics. They can be used to non-destructively constrain porosity as well as the geometry of pores and their distribution inside the sample (Svergun et al., 1987; Stock, 1999; Lindquist et al., 2000). X-ray computed tomography (CT) is the most readily available three-dimensional imaging method in geoscience. Two major sources of X-ray radiation are used in micro-CT applications: synchrotrons emitting collimated, almost parallel monochromatic X-rays, and X-ray tubes producing fan- or cone-beam polychromatic X-rays.

The scope of our work is to assess the impact of various limitations of X-ray CT as well as image processing methods on porosity estimations from three-dimensional X-ray micro CT images. We focus on global single-threshold methods requiring no user-dependent input parameters. For benchmarking these methods, we use a suite of eight sandstone varieties covering a large range in porosity. Porosities determined from X-ray CT images using a range of sample sizes are compared to laboratory measurements. In addition, we performed conventional stereological analysis of SEM images and examined thin-sections.

## 2 X-ray computed tomography

X-ray computed tomography (CT) is a non-destructive three-dimensional imaging technology. The method is based on the attenuation X-rays experience when passing through a solid object, such as a rock sample. The attenuation of the X-rays is quantified by the *Lambert-Beer law* (Beer, 1852)

$$I = I_0 e^{L\mu},\tag{1}$$

where $I_0$, $I$, $\mu$ and $L$ denote initial intensity, the intensity after the X-rays passed through an object, the linear attenuation coefficient and the thickness of the object, respectively. Attenuation is caused by processes such as photoelectric absorption or Compton scattering and therefore the X-ray attenuation coefficient is related to the atomic number of a material. Thus, X-ray attenuation can be used to image density contrasts.

Decades after first pioneering applications of X-ray in geoscience (Calvert and Veevers, 1962; Wellington and Vinegar, 1987; Coles et al., 1991) porosity determination using tomographic images is still a challenge, as one has to reliably identify and separate void space and solid phases in the image data. Various inherent physical and technical limitations complicate the evaluation of X-ray CT images (see e.g. Clausnitzer and Hopmans, 2000; Ketcham and Carlson, 2001; Cnudde and Boone,



2013; Wildenschild and Sheppard, 2013, for reviews). These limitations include partial-volume effects, instrumental noise, reconstruction artifacts and effects due to the use of polychromatic X-rays.

The distribution of X-ray attenuation in a sample is computed by combining a number of projection images taken from various angles and using inversion algorithms, such as filtered back-projection, to reconstruct a volume image (Kak and Slaney,

2001). The result is a discretized volume consisting of small volume elements or *voxels*. An intensity value representing the local X-ray attenuation coefficient is computed for each voxel. Thus, X-ray attenuation coefficients are averaged over small discrete volume elements. A homogeneous material or a mixture of heterogeneous features significantly smaller than the voxel size might therefore yield a voxel with the same intensity. This non-uniqueness is called the partial-volume (PV) effect, which results in a blurring of all phase boundaries in a CT image. The spatial resolution of the tomographic image produced by

conventional X-ray CT is limited by the diameter of the X-ray source, the X-ray photon energy, the absorption of the sample as well as the number and size of pixels on the detector. The nominal spatial resolution is usually defined from measurements of the modulation transfer function or from imaging a series of stationary phantoms e.g. lattices with sharp contrast differences and decreasing size under ideal conditions (e.g. Bouxsein et al., 2010). However, the (isotropic) voxel size is the calculated size of a discrete cubic volume element in the reconstructed image, which can actually be lower than the actual spatial resolution

of the image. Still we use the isotropic voxel size as resolution measure because the value is independent of system factors, such as detector noise and reconstruction algorithm. The term resolution is gaining significance, when set into relation to the size of features of interest, in our case the pore size. A resolution higher than the size of pores is mandatory a prerequisite for resolving features accurately.

Limitations for the accuracy of X-ray CT images also arise from beam fluctuations or off-focal radiation (Boone et al., 2012),

non-trivial interactions of (polychromatic) X-rays with the sample material, such as beam hardening (van de Casteele et al., 2002), or scattered X-rays hitting the detector (Glover, 1982). Furthermore, projection images are recorded using detectors with finite signal-to-noise ratios and manufacturing imperfections lead to non-linear and/or non-uniform pixel response. Any X-ray detection system therefore inherently adds some random and/or systematic variation to the actual signal. In the reconstructed volume data, these variations manifest as noise, ring artifacts (Sijbers and Postnov, 2004) or streak artifacts and halos around

high density material (de Man et al., 1998). These artifacts are often hard to distinguish from real features in the sample and therefore complicate a proper identification of phases (Jovanović et al., 2013).

Standard CT systems do not record voxel intensities in terms of any physical unit but rather map the data to a somewhat arbitrary range meeting either the data storage requirements (e.g. 0 to 65535 for common 16-bit integer dataformats) or, to ensure compatibility of images from different CT systems, transforming the data to so-called Hounsfield units or CT numbers.

The latter relate the computed intensity values to the attenuation of distilled water as well as air to make CT images from different CT systems comparable (Levi et al., 1982). For visualization purposes, the voxel intensity is usually displayed in gray-values with light and dark colors representing more and less attenuating material, respectively.

A straightforward and commonly used method for the identification and separation of phases in CT images – a process also called *segmentation* – is based on assigning non-overlapping ranges of intensity values to individual phases, i.e., using

global threshold values. The boundaries of these ranges are either determined by visual inspection (e.g. Leu et al., 2014) or



automatically using statistical information from the image data (Sezgin and Sankur, 2004b; Iassonov et al., 2009; Wildenschild and Sheppard, 2013). Due to the non-physical units of the image data, this process has to be applied to every image individually. For porosity estimations, usually only two phases are considered – solid material and pore space – separated by a single threshold value.

A number of more complex methods for phase separation incorporate multiple or local threshold values based on e.g. spatial information or gradients of the intensity distribution such as the watershed algorithm (Vincent and Soille, 1991) or converging active contours (Sheppard et al., 2004). To account for the partial-volume effect and sub-resolution porosity, some authors assign multiple thresholds to phases and their mixtures. Then voxels are identified as "sub-voxel porosity" in addition to "pore" and "solid" (Sok et al., 2010; Teles et al., 2016). Advanced image-acquisition techniques, such as dual energy X-ray CT (Alves

et al., 2015; Teles et al., 2016) or injection of contrast agents into the sample and multiple scanning (Akin and Kovscek, 2003; Klobes et al., 1997; Lu et al., 2000; Mayo et al., 2015), are used to enhance the phase contrast. The latter technique requires a differential imaging or so called registration and subtraction procedure. In addition, synchrotron radiation facilities offer high resolution, fast image acquisition of a few nanometer resolution, but are usually restricted to small sample sizes in the range of tens of microns.

## 3   Material and methods

We selected eight sandstone (sst) varieties representing a wide range in composition as well as grain and pore sizes (Table 1). The set comprises well-characterized, rather homogeneous, and readily available rock types, including Berea sst, Bentheim sst, Crab Orchard sst, Fontainebleau sst, Oberster sst, Pfaelzer sst, Ruhr sst, and Wilkeson sst, mostly devoid of significant macroscopic evidence for bedding or anisotropy. Three cylindrical cores with diameters of about 3, 10 and 30 mm were

diamond-drilled from each of the ten sandstone blocks including three different Fontainebleau varieties. The samples were cut and precision ground to a length-to-diameter ratio of about 2.5.

### 3.1   Petrographic and microstructural sample characterization

For petrographic and microstructural characterization, we analyzed polished thin sections (Figure A1) of all samples using plane and cross-polarized light microscopy as well as scanning electron microscopy, the latter after carbon coating the thin-

sections (Figure 1). Backscattered electron (BSE) images were captured with a Zeiss LEO 1530 Gemini field emission scanning electron microscope (SEM). To constrain porosity we performed point counting (Carver, 1971) on an area of 6.6 mm$^2$ of the SEM images, which included 130 measurement points. The measurements were repeated ten times at several locations. We determined the volumetric fractions of the major constituents from counting at least 1000 points on a regular grid under cross-polarized light. The results presented in Table 1 assume that the samples are isotropic and no additional biases such as plug-outs

are introduced during the preparation process.

    Quartz is the main constituent of all our sandstone samples. Quartz content ranges from less than 50 % for Wilkeson sst to almost 100 % for Fontainebleau sst (Figure 2). Feldspars and clay minerals are the most important secondary components

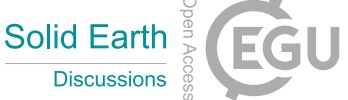



**Table 1.** Petrological and microstructural properties of the selected sandstone varieties.

| sandstone variety | label | mineralogical composition (%) | | | | grain shape | grain size (µm) | degree of cementation | ratio matrix/ cement |
|---|---|---|---|---|---|---|---|---|---|
| | | quartz | f-spar | clay | others | | | | |
| Berea | B | 82.1 | 6.1 | 5.0 | < 10 | sub-rounded | 150 | low | 5.8 |
| Crab Orchard | COB | 74.7 | < 3 | 17.4 | 5 | sub-angular | 250 | low | 3.4 |
| Fontainebleau | FS 8 | < 99 | < 1 | - | < 1 | sub-angular | 150 - 250 | high | 4.4 |
| Fontainebleau | FS 9 | < 99 | < 1 | - | < 1 | sub-angular | 150 - 250 | medium | 4.4 |
| Fontainebleau | FS 11 | < 99 | < 1 | - | < 1 | sub-angular | 150 - 250 | low | 4.4 |
| Bentheim | GBS | 95.3 | 4.3 | - | < 1 | sub-rounded | 250 | low | 10.2 |
| Ruhr | GR | 57 | 6.6 | 30.1 | ca. 5 | sub-angular | 100 - 200 | high | 30.1 |
| Oberster | OS | 73.4 | 5.8 | 5.9 | < 15 | sub-angular | 200 | high | 9.6 |
| Pfaelzer | PS | 71.7 | 11.0 | 2.9 | < 15 | sub-angular/angular | 150 | low | 7.2 |
| Wilkeson | WS | 47.6 | 20.6 | 21.8 | < 10 | sub-rounded | 200 | medium | 20.3 |

with highly variable abundance. Clay minerals, mainly a product of sericitization, make up to 30 % of Ruhr sst but are almost absent in Fontainebleau sst. Additional minor minerals with abundances of up to 5 % are muscovite, illite, chlorite, dolomite or calcite, and iron oxides. Accessory minerals such as rutile or zircon are present as well. For a detailed description of the different lithologies please refer to Appendix A.

5  The ten SEM images of each sample with a resolution of 2.3 µm per pixel were used to estimate semi-quantitatively the void size distribution by determining the equivalent void diameters $d_{\mathrm{eqv}}$ according to

$$d_{\mathrm{eqv}} = \sqrt{4A/\pi}, \tag{2}$$

where $A$ denotes the void area. For the calculation, the SEM images were automatically segmented into binary images using the method by Otsu (1979) to determine a global threshold (see Section 3.3.4). Continuous areas of pixels with the void characteristics are labeled and for each of these features (2) is used. We refrained from separating the labeled voids into specific pores and throats for instance by applying a watershed algorithm (e.g. Rabbani et al., 2014). Thus, our void-size distributions provide a suitable basis for the assessment of feature size rather than a precise characterization of the pore space geometry.





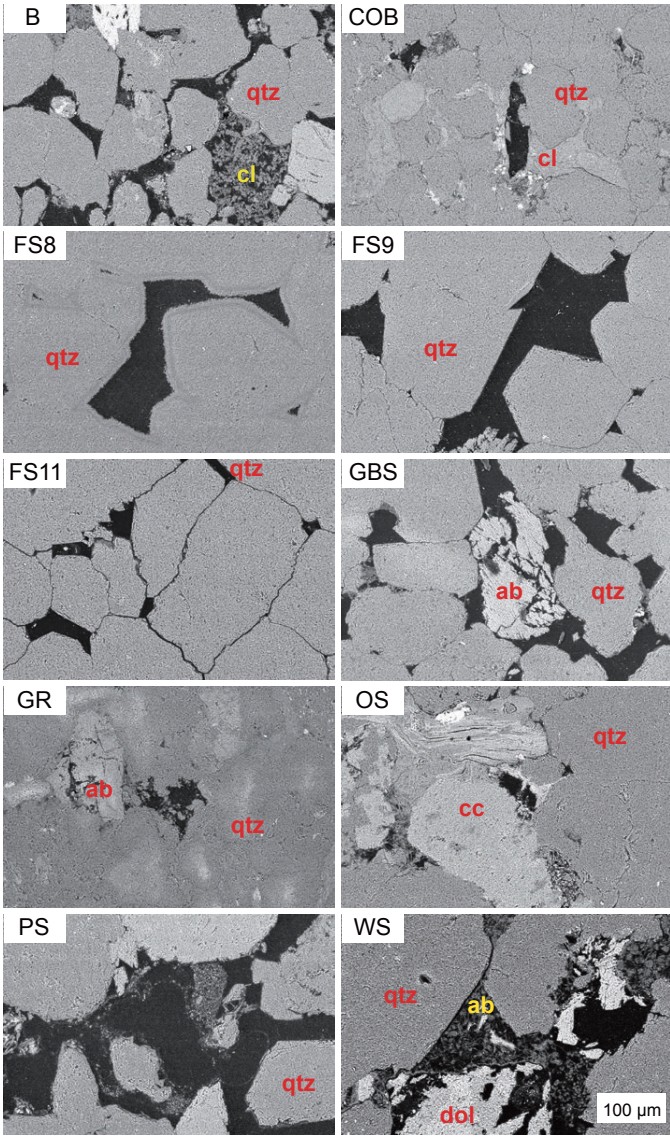

**Figure 1.** SEM micrographs of Berea (B), Crab Orchard (COB), Fontainebleau (FS), Bentheim (GBS), Ruhr (GR), Oberster (OS), Wilkeson (WS) and Pfaelzer (PS) sandstone. Minerals indicated are albite (ab), ankerite (ak), calcite (cc), clay minerals (cl), dolomite (dol) and quartz (qtz). Secondary quartz fringes document cementation in sample FS8.





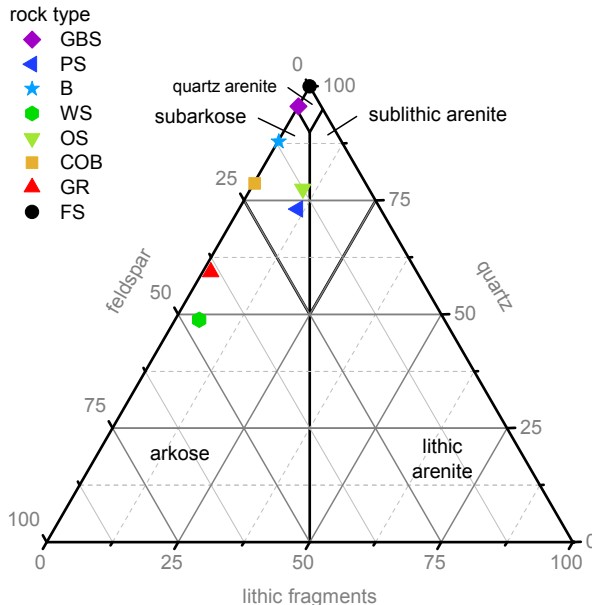

**Figure 2.** Classification of our sample selection following Dott (1964), but ignoring the clay content. The examined sandstones are mainly comprised of quartz and feldspar and have minor lithic fragments.

## 3.2 Laboratory measurements of porosity

We determined two measures of sample porosity, total porosity and effective porosity. The total porosity $\phi_{\mathrm{tot,lab}}$ is calculated from the ratio between bulk density $\rho_{\mathrm{b}}$ and matrix density $\rho_{\mathrm{m}}$ according to

$$\phi_{\mathrm{tot,lab}} = 1 - \frac{\rho_{\mathrm{b}}}{\rho_{\mathrm{m}}}. \tag{3}$$

5  Bulk density was derived from the samples with 30 mm diameter by weighing after drying at 60°C and geometrical estimation of bulk volume $V_{\mathrm{b}}$ according to

$$\rho_{\mathrm{b}} = \frac{m}{V_{\mathrm{b}}} = \frac{m}{\pi r^2 h}, \tag{4}$$

where $d$ denotes the sample diameter and $h$ the sample height. Matrix or grain density $\rho_{\mathrm{m}}$ was derived from measurements with a capillary pycnometer according to DIN 18124 using powders prepared by crushing and grinding pieces of the sandstone

10  blocks and distilled water as working fluid:

$$\rho_{\mathrm{m}} = \frac{m_{\mathrm{powder}}(m_{\mathrm{pyc+fl}} - m_{\mathrm{pyc}})}{V_{\mathrm{pyc}}(m_{\mathrm{pyc+fl}} - m_{\mathrm{pyc+fl+grain}} + m_{\mathrm{powder}})}, \tag{5}$$

where $m_{\mathrm{powder}}$, $m_{\mathrm{pyc}}$, $m_{\mathrm{pyc+fl}}$, $m_{\mathrm{pyc+fl+grain}}$ and $V_{\mathrm{pyc}}$ denote the masses of crushed sample powder, empty pycnometer, mass of pycnometer filled with water, mass of pycnometer filled with water and powder and the volume of the pycnometer, respectively. We report the average of three measurements for bulk and matrix densities.





Effective porosity $\phi_{\mathrm{eff,imb}}$ was deduced from the mass gain due to water imbibition. We evacuated and then immersed the samples in deionized water for at least 24 hours. We measured the water content by weighing samples after removing excess water from their surfaces. The effective porosity $\phi_{\mathrm{eff,imb}}$ calculates from the difference in mass before ($m_{\mathrm{dry}}$) and after saturation ($m_{\mathrm{sat}}$) according to

$$\phi_{\mathrm{eff,imb}} = \frac{m_{\mathrm{sat}} - m_{\mathrm{dry}}}{\rho_{\mathrm{H_2O}} V_{\mathrm{b}}}. \tag{6}$$

We also determined the effective porosity using the gas expansion method (Washburn and Bunting, 1922). First samples were placed in a pressure chamber of volume $V_2$ and the system was evacuated reaching a pressure level $p_{2,\mathrm{in}}$ close to $10^2$ Pa. Pressurized Argon was then released from a second chamber of volume $V_1$ with a pressure $p_{1,\mathrm{in}}$ of around 1 MPa. After expansion of the gas, a pressure level $p_{\mathrm{final}}$ is reached in the two connected chambers. Employing the ideal gas law, the effective matrix volume $V_{\mathrm{eff,matrix}}$ is calculated according to

$$V_{\mathrm{eff,matrix}} = -V_1 - V_2 + V_1 \frac{p_{1,\mathrm{in}} - p_{2,\mathrm{in}}}{p_{\mathrm{final}}}. \tag{7}$$

The effective porosity then is

$$\phi_{\mathrm{eff,gas}} = \frac{1 - V_{\mathrm{eff,matrix}}}{V_{\mathrm{b}}}. \tag{8}$$

To assess the quality of our porosity measurements we calculated the propagation of uncertainties (Taylor, 1997). A detailed description of the uncertainties for our laboratory measurements is given in Appendix A2.

## 3.3 Quantification of porosity from CT images

### 3.3.1 X-ray computed tomography

The tomographic datasets were recorded with a CT scanner manufactured by ProCon X-Ray based on their CT-Alpha model. The CT scanner is equipped with a multifocal X-ray tube with a maximum acceleration voltage of 225 kV and with a maximum electron-beam power on target of 2 W, 5 W and 50 W for spot diameters of 1 µm, 5 µm, and 25 µm, respectively. The transmitted X-ray intensities are captured using a 190 by 240 mm Varian flat-panel detector with 1516 by 1900 pixels. Each pixel in the detector has an edge length of 127 µm and a gray-value resolution of 14 bit resulting in a nominal system resolution of around 1 µm. In our set-up, the X-ray source and detector are fixed during imaging process, while the sample is continuously rotated. We recorded a total of 1600 images corresponding to rotation steps of 0.2°.

The sample diameter determines the scanning parameters. We positioned the sample and detector in such a way, that the projection of the object almost completely covers the detector. The resolution is controlled by sample position between X-ray tube and detector. The geometrical magnification

$$M = \frac{SDD}{SOD} \tag{9}$$

of an X-ray system is defined by the source-detector-distance (SDD) divided by the source-object-distance (SOD) (Voland et al., 2010). Consequently, the volume resolution (voxel size) increases with decreasing sample diameter of the cylindrical




cores as a function of the geometric magnification and the number and size of the detector pixels. Cores with a diameter of 30 mm correspond to a voxel size of about 21.8 µm, while those with 3 mm yield the highest resolution with voxel sizes of about 2.5 µm (Table B1).

We adapted acceleration voltage, current and exposure times to make optimal use of the detector sensitivity. The measure-
ments were performed at acceleration voltages between 70 to 160 kV and target currents between 25 to 180 µA depending on core size and material (Table B). We implemented metal filters that fit exactly on the collimator cover of the X-ray tube to pre-harden the X-rays and to reduce beam hardening artifacts. We recorded the majority of samples with 3 and 10 mm diameter using an aluminum filter of 1 mm thickness. A 0.5 mm thick copper filter was implemented for all samples with 30 mm diameter. Exposure times were 100 ms, 300 ms, and 450 ms for 3 mm, 10 mm, and 30 mm samples, respectively. Some 3 mm
samples were scanned with a focal spot diameter of 1 µm at 80 kV and 25 µA. To counterbalance the reduced beam power, we then increased the exposure time to 1000 ms and omitted the pre-hardening of the X-rays.

### 3.3.2 Reconstruction and image processing

The reconstruction process for the conical X-ray beam, based on a Radon transform in the form of a convolution and back filter (Feldkamp et al., 1984; Turbell, 2001), was performed with the Volex reconstruction software package (Fraunhofer-Allianz
Vision, 2012). The software also corrects for ring-artifacts. The voxel intensities are derived from the loss of energy of the X-rays and are mapped to a 16-bit integer range between zero, corresponding to the background, in our case air, and 65535, corresponding to the detection limits of the CT-Scanner.

After the 3D volume reconstruction, we processed and filtered the images using the software package *Avizo Fire, Version 9.0.1* [©] (FEI Visualization Sciences Group, 2016). In a first step, we divided the reconstructed datasets into nine cuboid
subvolumes with a size of 350×400×500 voxels located well inside the sample volume to reduce computation time and to exclude non-representative surface areas and avoid the rims affected most by beam hardening artifacts. We selected the subvolumes to overlap by 50 voxels along one edge (Figure 3).

We tested the influence of image filtering on segmentation results by processing four different datasets: unfiltered data and data filtered with a median filter (Median), a non-local means filter (NLM), and a combination of both (Median+NLM). The
median filter is a morphological filter that replaces a specific entry of the data matrix with the median of specified neighboring entries (Huang et al., 1979; Ohser and Schladitz, 2009). We applied the median filter in three dimensions, comparing each voxel with 26 neighbor voxels by using a 3x3x3 cube as structuring element. We applied the filter in three successions. Apart from random noise, the filter supposedly also reduces ring artifacts originating from the sample's rotation during the scanning process (Ketcham and Carlson, 2001).
The NLM filter employs an algorithm in which a voxel of interest is weighted by the similarity to its surrounding voxels (Buades et al., 2005). We used two different windows, with a diameter of 21 voxels for the determination of the similarity weights and a diameter of 5 voxels for the actual weighted median filtering.




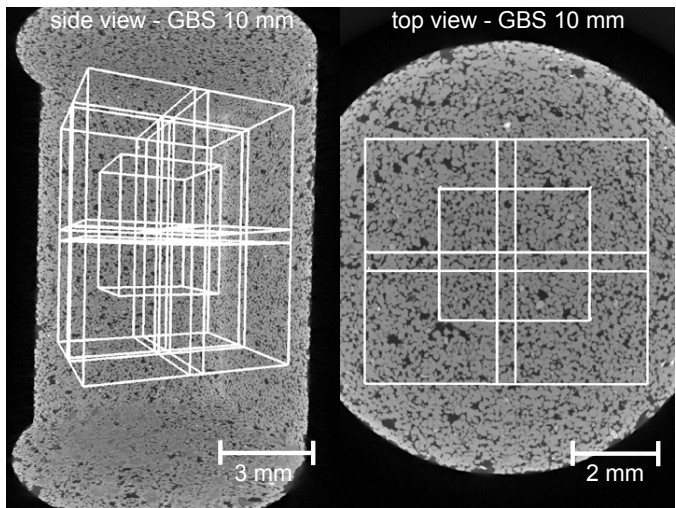

**Figure 3.** Example of subvolume selection from the raw datasets of the Bentheim (GBS) sandstone (GBS) sample with 10 mm diameter in x-z (left) and (right) z-direction parallel to core-axis.

### 3.3.3 Image segmentation and porosity determination

To visualize the distribution of voxel intensities in the image data, we use histograms that represent the frequency of each intensity value of the reconstructed image data. The histogram thus provides a general overview of the image characteristics, such as gray-level distribution, which is a function of the material densities, the average luminance of the images as well as

contrast and noise. The intensity distributions can be described using statistical terms including *mean*, which is the average intensity, *median*, which divides all intensity values into two equal halves, and *mode*, which is the most frequently occurring value. When these three measures fall on the same intensity value, the intensity distribution is a *normal distribution*. In contrast the mean and median are not identical for asymmetrical distributions, which are called *skewed*. For a dense mono-mineralic, continuous and homogeneous material we would expect a more or less normal distribution of intensities due to image noise.

When several of such materials compose a sample, we might be able to identify several *classes* of voxel intensities by the occurrence of local maxima, i.e., mulitmodal distributions. These classes can be overlapping yielding skewed distributions and complicating separation of the classes.

For a quantitative characterization of pore space, the intensity image is converted into a binary image. In this binary image all voxels are either assigned to pore or solid (Iassonov et al., 2009; Andrä et al., 2013). This process is called image segmentation.

Thus, quantitative estimates of phase contents rely on an appropriate segmentation of the phases of interest (Stock, 2013). Generally, one distinguishes global and local segmentation methods (Sezgin and Sankur, 2004a; Kaestner et al., 2008; Iassonov et al., 2009; Schlüter et al., 2014). Global segmentation methods define absolute thresholds for the entire image. An initial





image $Img_{\mathrm{ini}}$, containing the intensity field $I_{\mathrm{ini}}(x,y,z)$, is converted to a segmented or binarized image

$$I_{\mathrm{bin}}(x,y,z) = \begin{cases} 0 \text{ if } I_{\mathrm{ini}}(x,y,z) < T \\ 1 \text{ if } I_{\mathrm{ini}}(x,y,z) \geq T \end{cases}, \tag{10}$$

with the threshold $T$ at a certain voxel intensity. This approach is easily extended to multiple classes by defining multiple thresholds. Thresholds are either determined - albeit somewhat arbitrarily - by visual inspection or by algorithms that are in

most cases based on statistical or geometrical evaluation of the image data histogram. In contrast, local segmentation accounts for neighborhood characteristics usually in combination with intensity thresholding as a starting point.

### 3.3.4   Implemented thresholding algorithms

We apply seven common global segmentation methods to each of the nine subvolumes extracted from the tomographic datasets. Binarization was carried out for unfiltered and filtered images. Some of the algorithms are designed for bimodal distributions,

while others are applicable to multimodal threshold problems with n classes and n-1 thresholds. We expect two distinct classes in the intensity distributions according to the significant density contrast of air with $1.2 \, \mathrm{kg/m^3}$ and rock forming minerals e.g. quartz with a density of about $2500 \, \mathrm{kg/m^3}$ (Deer et al., 2011). Open pore space should always be associated with lower intensity values than the minerals composing the investigated rocks. We therefore consider all voxels with intensities below the threshold as pore space and all voxels with higher intensities as solid material. In some samples with significant amount of high density

minerals, e.g. oxides and sulfides, the intensity classes of these minerals were much more prominent than the porosity class. Thus, to simplify thresholding, we ignored intensities larger than the intensity with the largest frequency $g_{\mathrm{mode1}}$, which we attribute to solid quartz because quartz is the major constituent of the examined sandstones.

#### Fixed ratio thresholding

For fixed ratio (FR) thresholding, we assume that an optimum threshold is fixed relative to the intensity of (quartz) grains (i.e.

$g_{\mathrm{mode1}}$) and the intensity level $g_{\mathrm{dark}}$ separating solid and pore space. We somewhat arbitrarily define the background intensity to be smaller than 0.001 times the counts of $g_{\mathrm{mode1}}$. The intensity level $g_{\mathrm{dark}}$ is then placed at the first intensity value exceeding the background intensities and which can not be attributed to the pore space. The FR algorithm sets the threshold between $g_{\mathrm{dark}}$ and $g_{\mathrm{mode1}}$ using a user-defined ratio. From a visual comparison of binarized images and the intensity images, we choose a threshold level of

$$T = g_{\mathrm{dark}} + \frac{3}{4}(g_{\mathrm{mode1}} - g_{\mathrm{dark}}). \tag{11}$$

#### Maximum variance

The maximum variance (MV) algorithm proposed by Otsu (1979) searches for the threshold $T$ that minimizes the variance in each class while maximizing the variance between the classes. For two classes, the intra-class variance $\sigma_w^2$ is defined as a



weighted sum of the variances $\sigma_0^2$ and $\sigma_1^2$ within the two classes:

$$\sigma_w^2 = \omega_0 \sigma_0^2 + \omega_1 \sigma_1^2. \tag{12}$$

The weighting factors $\omega_0$ and $\omega_1$ are the probabilities of the two classes separated by the threshold level. The variance between the classes calculates

$$\sigma_B^2 = \omega_0(\mu_0 - \mu_T)^2 + \omega_1(\mu_1 - \mu_T)^2, \tag{13}$$

where $\mu_T$ denotes the average of all values and $\mu_0$ and $\mu_1$ denote the mean values within each class defined. The algorithm maximizes

$$f(T) = \frac{\sigma_w^2(T)}{\sigma_w^2 + \sigma_B^2} \tag{14}$$

to find the threshold. We applied the MV thresholding algorithm only to the intensity range between zero and $g_{\mathrm{mode1}}$ to avoid misclassification by threshold levels higher than $g_{\mathrm{mode1}}$ in samples with significant volume fractions of high gray-value (bright) voxels.

**Unimodal thresholding**

Unimodal (UM) thresholding was designed for images with a unimodal intensity distribution (Rosin, 2001), i.e., where the intensity values of grains and pores substantially overlap in the histogram. The minor secondary population corresponds to the broad, gentle slope in the skewed intensity histogram. A straight line is drawn from the bin with the highest intensity to the first bin above the background noise level $g_{\mathrm{dark}}$. The threshold is then determined by finding the maximum perpendicular distance between this straight line and the histogram curve.

**Minimum error thresholding**

Minimum error (ME) thresholding introduced by Kittler and Illingworth (1986) assumes that the histogram is composed of normal distributions, i.e. Gaussian modes with mean $\mu_i$ and standard deviation $\sigma_i$ (Drobchenko et al., 2011). The criterion function

$$f(T) = 1 + 2[P_1(T)\log\sigma_1(T) + P_2(T)\log\sigma_2(T)] - 2[P_1(T)\log P_1(T) + P_2(T)\log P_2(T)]. \tag{15}$$

is minimized.

**Iterative k-means clustering**

Like the ME, the iterative k-means clustering (KM) (Ridler and Calvard, 1978) assumes Gaussian modes albeit with uniform standard deviations for every class (Birchfield, 2016). The algorithm partitions the voxels of an image into clusters by using a similarity feature, like intensity of voxels and/or distance of voxel intensities from centroids (Luo et al., 2014; Jaroš et al.,




2017). An arbitrary initial threshold, e.g. 50 % of the maximum intensity, defines the initial classes. Then a new threshold $T$ is determined according to

$$f(T) = \frac{\mu_0 + \mu_1}{2}, \tag{16}$$

where $\mu_0$ and $\mu_1$ denote the mean values of the two classes defined by the initial threshold (Ridler and Calvard, 1978). The
optimal threshold is calculated by iteratively repeating this step until the threshold level does not change any more.

**Shape thresholding and mid-point thresholding**

Shape (SH) thresholding by Tsai (1995) is a histogram-based approach for the segmentation of multiple phases in multimodal or unimodal histograms. The original histogram is first smoothed with a Gaussian window function, then local peaks and valleys are employed as thresholds. For unimodal histograms, the algorithm detects and selects the point of maximum curvature as
threshold level. However, for many of the unimodal histograms recorded here the algorithm selected the point of maximum curvature at a threshold level with an intensity larger than $g_{\mathrm{mode1}}$. Therefore, it failed to separate pore intensities from grain intensities and we did not employ this algorithm. Instead we slightly modified the SH algorithm using the mid-point (MP) thresholding method of Saenger et al. (2016). While the position of the major and minor mode of a bimodal histogram are determined as described above for the SH algorithm, a threshold $T$ is determined according to

$$T = g_{\mathrm{mode2}} + \frac{1}{2}(g_{\mathrm{mode1}} - g_{\mathrm{mode2}}). \tag{17}$$

## 4 Results

### 4.1 Laboratory measurements of porosity

Bulk densities of the investigated sandstones range from 2000 to 2600 kg m$^{-3}$, while grain densities range from 2600 to
2680 kg m$^{-3}$ (Table 2). Bentheim sst exhibits the lowest and Ruhr sst the highest bulk density of the sample collection. The
uncertainty is in the same range as the standard deviation for the three repeat-measurements of bulk density, while the uncertainty is two orders of magnitude lower than the standard deviation for grain density. The mismatch between uncertainty and standard deviation in case of grain density might indicate user-dependent inaccuracies in the experimental procedure such as unrecorded changes in the environmental conditions or insufficient evacuation of the pycnometers, due to e.g. clumping of sample powder.
Total porosity $\phi_{\mathrm{tot,lab}}$ covers a range of about 3 to 25 % (Figure 4, Table 2). In most cases, effective porosity $\phi_{\mathrm{eff,imb}}$ derived from water saturation and effective porosity $\phi_{\mathrm{eff,gas}}$ derived from gas pycnometer measurements either match total porosity $\phi_{\mathrm{tot,lab}}$ within uncertainty or fall below it (Figure 4). The effective porosities $\phi_{\mathrm{eff,gas}}$ deduced from gas pycnometer measurements are greater than the effective porosity $\phi_{\mathrm{eff,imb}}$ from water imbibition except for FS9. Standard deviations for several measurements are usually between 0.01 and 0.5 % for total porosity while error propagation yields absolute errors of $\pm 1.9$ %.
Total porosity derived from analyzing SEM images, $\phi_{\mathrm{tot,SEM}}$, shows large deviations compared to $\phi_{\mathrm{tot,lab}}$ and generally also




greater standard deviations (Table B1). The standard deviations, deduced from analyzing ten randomly selected images of each sample, range from 0.7 to 4.6 %. Based on the results of the laboratory measurements we classify the investigated sandstones in three groups with respect to porosity and mineralogical composition:

A. Highly porous (15 - 25 % porosity), quartz dominated (> 75 % quartz) sandstones: B, GBS and PS.

B. The suite of Fontainebleau sandstone samples with intermediate porosity (5 - 15 %) and quartz as main constituent (> 90 %): FS8, FS9 and FS11.

C. Tight sandstones with porosities below 10 % and variable mineralogy (clay content up to 20 %, quartz < 75 %): WS, COB, OS and GR, the latter representing an extreme case within this group with a clay content above 20 % and a porosity below 5 %.

The classification of the sandstone varieties into three groups is confirmed by the semi-quantitative pore-size distributions derived from analysis of SEM images (Figure 5). The mean equivalent pore diameters for highly porous sandstones are larger than 30 μm and reach values above 200 μm. The distributions of the Fontainebleau varieties resemble each other irrespective of total porosity $\phi_{\mathrm{tot,lab}}$. The mean diameters for rocks of group C fall significantly below the lowest resolution of about 22 μm used in the CT measurements at a sample diameter of 30 mm. The amount of 211 designated pores for GR is approximately
one order of magnitude lower than for the other examined rocks.





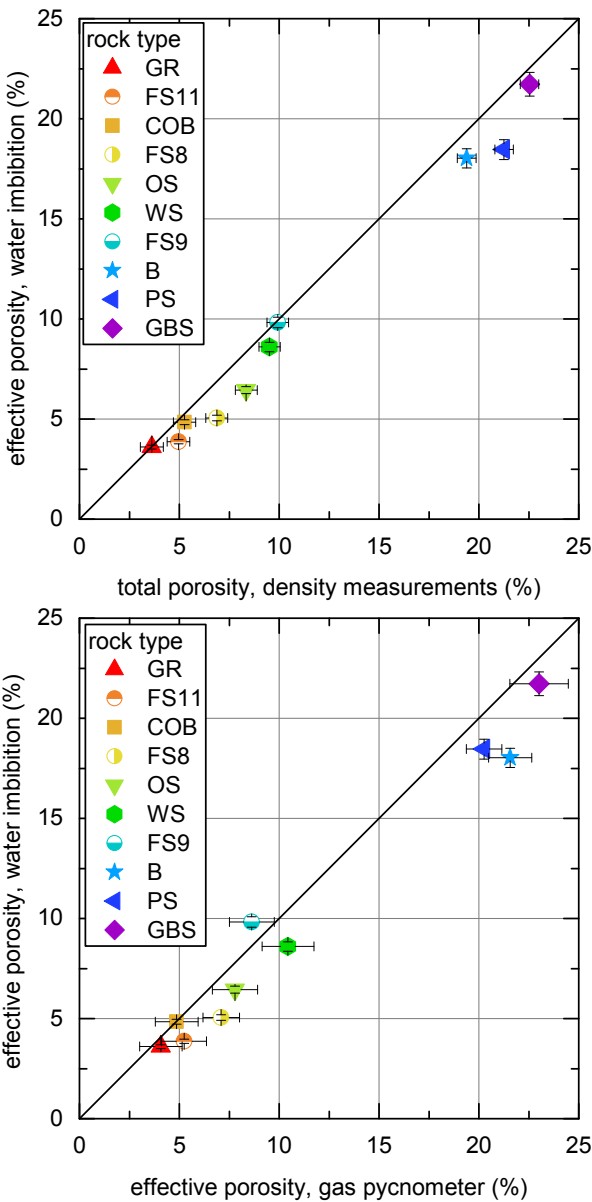

**Figure 4.** Correlation between a) experimentally determined total porosity ($\phi_{\mathrm{tot,lab}}$) and effective porosity ($\phi_{\mathrm{eff,imb}}$) from water imbibition and b.) effective porosity ($\phi_{\mathrm{eff,gas}}$) deduced from gas pycnometer measurements and the effective porosity ($\phi_{\mathrm{eff,imb}}$) through water imbibition. The straight gray lines indicate identity. The error bars give the experimental uncertainties.



**Figure 5.** Normalized distribution of equivalent pore diameters based on the evaluation of SEM images at a resolution of 2.3 µm. The voxel sizes of 2.5 µm, 8 µm and 22 µm are indicated by dashed lines to show their relation to the mean pore diameter indicated by a solid black line. Labels give the total porosity $\phi_{\mathrm{tot,lab}}$ from laboratory measurements and the amount $n$ of detected voids.



**Table 2.** Experimentally determined density and porosity of the sample suite including propagated uncertainties (labels as in table 1). The P-wave velocity measurements of dry samples are displayed in order to supplement the existing measures.

| sample | P-wave velocity | density | | total porosity | | effective porosity | |
|---|---|---|---|---|---|---|---|
| | | bulk | grain | $\phi_{\mathrm{tot,lab}}$ by by density | $\phi_{\mathrm{tot,SEM}}$ by point counting | $\phi_{\mathrm{eff,imb}}$ by imbibition | $\phi_{\mathrm{eff,gas}}$ by pycnometer |
| | km s$^{-1}$ | kg m$^{-3}$ | kg m$^{-3}$ | % | % | % | % |
| B | 2.58±0.04 | 2147.4±12.6 (±1.6)[*] | 2664.0±0.2 (±10.3)[*] | 19.39±0.47 | 14.44±4.60[*] | 18.02±0.48 | 21.56±1.08 |
| COB | 3.63±0.08 | 2535.8±14.9 (±5.6)[*] | 2676.5±0.2 (±13.6)[*] | 5.26±0.56 | 3.08±0.69[*] | 4.84±0.13 | 4.87±1.07 |
| FS8 | 5.46±0.14 | 2479.9±14.7 (±5.3)[*] | 2663.0±0.3 (±7.4)[*] | 6.88±0.55 | 5.69±2.76[*] | 5.06±0.14 | 7.09±0.91 |
| FS9 | 2.46±0.03 | 2380.4±14.3 (±25.2)[*] | 2642.8±0.6 (±1.3)[*] | 9.93±0.54 | 9.31±2.16[*] | 9.83±0.26 | 8.63±1.12 |
| FS11 | 2.01±0.02 | 2531.9±15.1 (±9.4)[*] | 2664.0±0.2 (±19.1)[*] | 4.96±0.57 | 6.38±2.04[*] | 3.87±0.10 | 5.24±1.13 |
| GBS | 2.86±0.05 | 2017.9±12.0 (±15.2)[*] | 2605.0±0.7 (±10.3)[*] | 22.54±0.46 | 1.77±1.09[*] | 21.72±0.59 | 23.01±1.46 |
| GR | 3.29±0.05 | 2585.2±15.4 (±5.2)[*] | 2682.5±0.1 (±6.7)[*] | 3.63±0.57 | 1.77±1.09[*] | 3.61±0.10 | 4.07±1.06 |
| OS | 4.03±0.09 | 2453.6±14.7 (±5.9)[*] | 2677.2±0.5 (±17.1)[*] | 8.35±0.55 | 3.94±2.20[*] | 6.44±0.17 | 7.78±1.13 |
| PS | 3.40±0.05 | 2111.9±12.3 (±3.8)[*] | 2682.0±0.3 (±37.3)[*] | 21.26±0.46 | 15.77±3.08[*] | 18.46±0.50 | 20.26±0.88 |
| WS | 2.87±0.03 | 2418.3±14.2 (±16.4)[*] | 2672.6±0.4 (±24.4)[*] | 9.52±0.53 | 6.15±2.09[*] | 8.60±0.07 | 10.44±1.29 |

[*] standard deviation

## 4.2 Voxel-intensity distribution in raw images

Images obtained using X-rays pre-hardened by filters have a higher signal to noise ratio than those taken without a filter. The quality of images taken with a focal spot diameter of 1 μm and without X-ray filters (Figure 6, FS9) is similar to images recorded with a focal spot diameter of 5 μm using aluminum filters (Figure 6, FS8).

**Figure 6.** Gray-scale slice images of the voxel intensities for each sandstone with voxel sizes of about 2.5 μm, 8 μm and 22 μm corresponding to the sample sizes of 3, 10 and 30 mm, respectively. Each image has an edge length of 400 voxels and absolute voxel intensities are displayed using a uniform gray value range. Porosity from laboratory measurements $\phi_{\mathrm{tot,lab}}$ is given below each sample name. The small inserts show histograms of the voxel intensities for each image.









Some general trends emerge from the intensity histograms of the reconstructed images (Figure 6). As all histograms in Figure 6 are scaled equally, higher average voxel intensities indicate higher signal to noise ratios. In general, images with a voxel size of 2.5 μm have a lower signal to noise ratio than images with lower resolutions due to reduced X-ray energies. The low-resolution images of FS yield the highest signal to noise ratios. Because the absolute voxel intensities are not normalized to a common range of values but depend on the CT system configuration, including sample size and X-ray beam energy, the absolute voxel intensities cannot be directly related to physical units.

We can distinguish four classes of voxel-intensity distributions for the images:

1. Distinctly bimodal distributions, which occur only for the high-resolution images of group A rocks.

2. Bimodal distributions with a negative skewness of the major peak and a rather small secondary minor peak. These distributions are characteristic for CT images of group A at intermediate resolution as well as high-resolution images of rock samples from group B.

3. Low resolution images of group A rocks as well as some intermediate resolution images of group B rocks yield unimodal distributions with negative skew and long tails.

4. All images of group C rocks as well as some low-resolution images of group B rocks correspond to unimodal intensity distributions without a distinct skewness.

Filtering generally alter these voxel-intensity histograms. The class modes increase and the widths at half maximum decrease (Figure 7). When the distributions of multiple classes overlap, filters separate these classes in some cases, i.e. single skewed peaks in the unfiltered data are then transformed to multiple peaks (Figure 7a,b). However, small classes as well as minor local maxima are usually smoothed out and disappear from the histograms. In addition, median filtering results in blurred phase boundaries in the images (Figure 7f). The combined Median+NLM filter yields nearly identical results to the median filter only.

Compared to the median filter, the NLM filtering increases the contrast at phase boundaries significantly. The filter removes statistical noise without considerably altering the geometry of small features (compare Figure 7e and 7g). Valleys between modes become more pronounced in the intensity histogram (Figure 7c).

The appearance of histograms, (1) distinct bimodal, (2) skewed bimodal with small secondary peak , (3) skewed unimodal and (4) symmetric unimodal depends on the resolution of the image as well as the texture and mineralogy of the sample (see Figure 8). A symmetrical peak appearance is characteristic for FS samples and GBS sample predominantly composed of quartz. The major mode is represented by a normal distribution of voxel intensities, which are attributed to solid grains i.e. quartz. The smaller, secondary peak in the bimodal intensity histograms is considered to represent the pore space. However, in most cases the different phases are not represented by well separated, normal distributed X-ray coefficient classes. Instead, the histograms represent a superposition of overlapping distributions. A distinction between the voxel intensities of pore space and solid grains is not straightforward for multi-phase samples like GR and PS characterized by a prominent skewness of the maximum peak to the right, which seems to reflect compositional variability. The significant amount of high intensity voxels



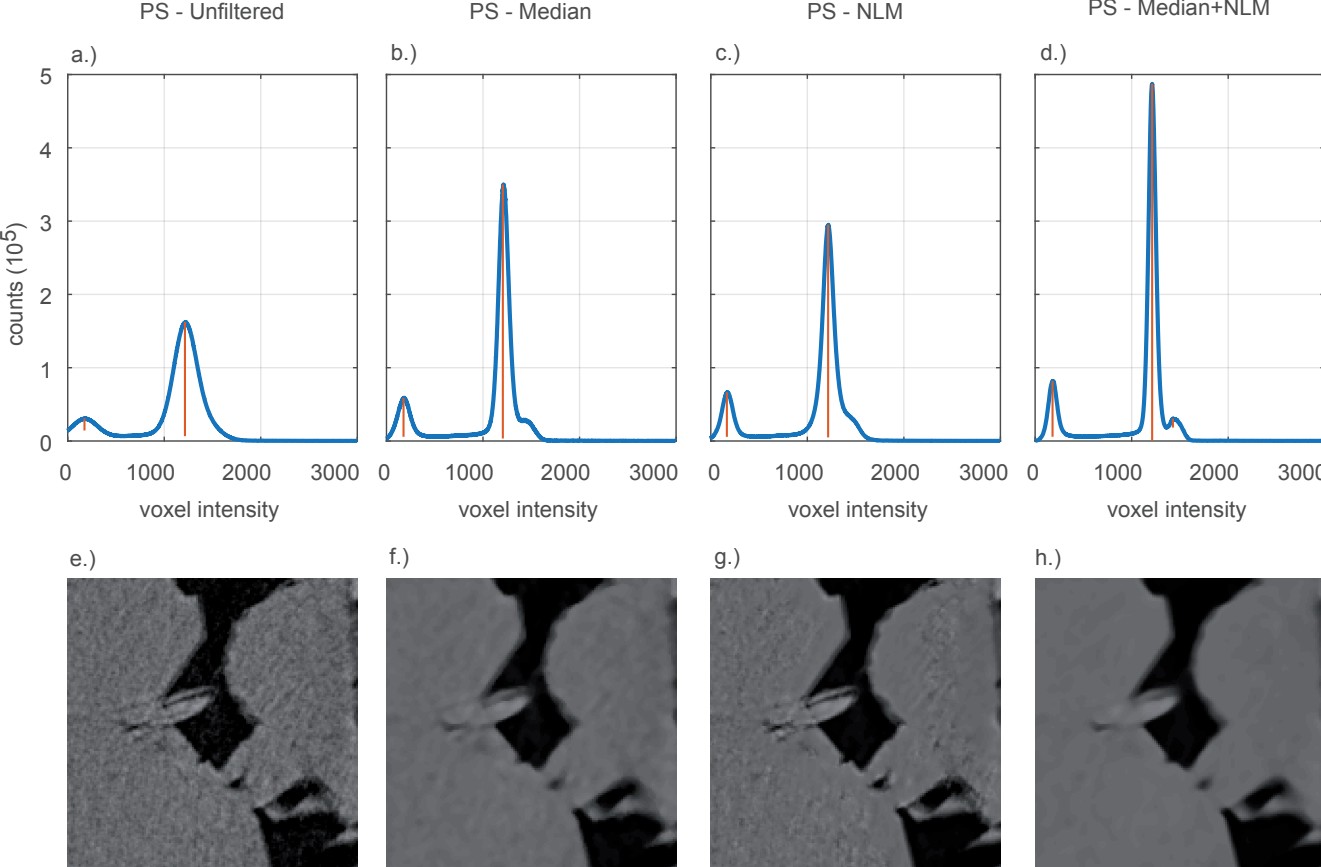

**Figure 7.** Comparison of unfiltered (a, e), median filtered (b, f), non-local means filtered (c, g) and consecutively median and non-local means filtered (d, h) voxel intensity histograms and corresponding images of the sample of Pfälzer sandstone (PS) with a diameter of 3 mm. Filter operations result in an increase in peak intensities for all filters we used. Vertical orange lines indicate the major modes or number of classes. Filtering reduces the speckled appearance of the solid phase (gray), but blurs the phase boundary to the pores (black).

in those images can be attributed to minerals with densities higher than quartz. These minerals are for example anorthite with a density of 2760 kg m$^{-3}$ (Smith, 1974), sericite with a density of around 2820 kg m$^{-3}$ (Deer et al., 2011), and metal oxides or sulfides. Skewness to the left side of the maximum peak however reflects insufficient separation of solid components and open voids due to e.g. void sizes that are small compared to the resolution.

5    The effect of the ratio between void size and resolution on the appearance of the intensity histogram is demonstrated by the series of measurements performed on the three differently sized samples of a specific sandstone variety. Analysis of SEM images yield a mean equivalent void diameter of about 50 µm for PS (Figure 1), representative for rocks of group A, and thus one order of magnitude above the resolution of 2.5 µm for samples with a diameter of 3 mm, while void diameter and resolution are similar for a sample diameter of 30 mm. The distribution changes from bimodal to skewed unimodal with increasing sample





size (Figure 8). The amount of pores larger than 5 µm is limited in the samples of group C resulting in unimodal distributions (Figure 8). Hence, at a voxel size of 2.5 µm a large amount of unresolved pore space is incorporated as partial-volume (PV) voxels in the reconstructed gray-value images.

### 4.3 Global thresholding and porosity calculation

For comparison, all thresholding algorithms were applied to the unfiltered and filtered images. Images with distinct bimodal voxel-intensity histograms consistently yield threshold levels in the valley between the two peaks in the histogram. All values beneath the threshold level are considered to represent pores (Figure 8). The various threshold levels scatter more for histograms of class 2, than for distinct bimodal distributions. For the unimodal histograms of class 3 and 4, thresholding algorithms place the threshold levels between the bend and the inflection point to the left of the major mode.

We calculated the mean value for total porosity $\phi_{\text{tot,CT}}$ and the corresponding standard deviations $\sigma_{\text{tot,CT}}$ by averaging over nine subvolumes for each of the applied thresholding methods (Figure 9). In most cases, porosity $\phi_{\text{tot,CT}}$ derived from the CT images significantly deviates from porosity determined in the laboratory (Figure 9). Moreover, the different thresholding algorithms yield highly variable porosity estimates. The discrepancy between laboratory measurements and CT-derived porosities is up to more than $\delta\phi = 20\,\%$. Images of GBS ($\phi_{\text{tot,lab}} = 22.5\,\%$) yield the highest variability in $\phi_{\text{tot,CT}}$ ranging from around

37 down to 2 % at resolutions of about 8 and 22 µm. Rocks of group C yield $\phi_{\text{tot,CT}}$ close to zero comparable to $\phi_{\text{tot,lab}}$ from laboratory measurements. The higher degree of secondary quartz cementation in the monomineralic samples of Fontainebleau FS8 compared to FS11 does not have any recognizable impact on porosity estimates.





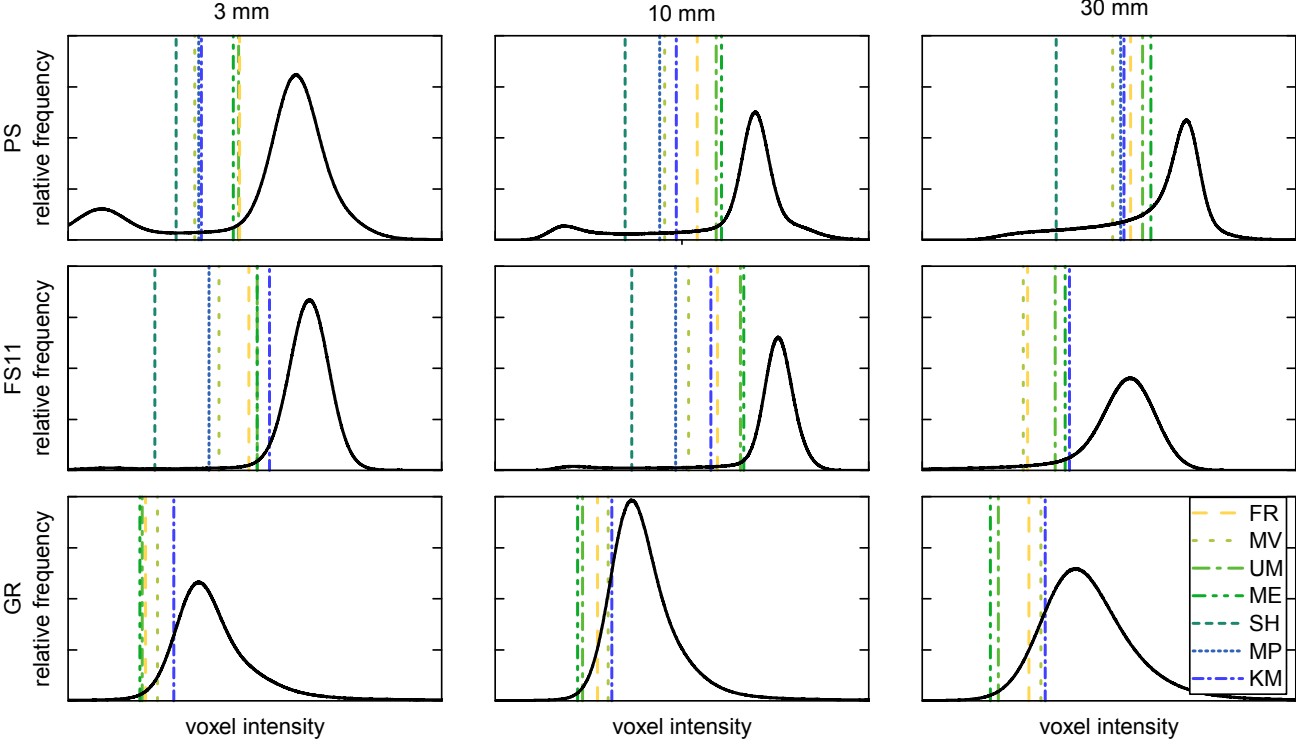

**Figure 8.** Relative voxel-intensity histograms with thresholding levels by global segmentation methods (FR) fixed ratio, (MV) maximum variance, (UM) unimodal, (ME) minimum error, (KM) k-means, (SH) shape and (MP) mid-point of the samples PS, FS11 and GR.



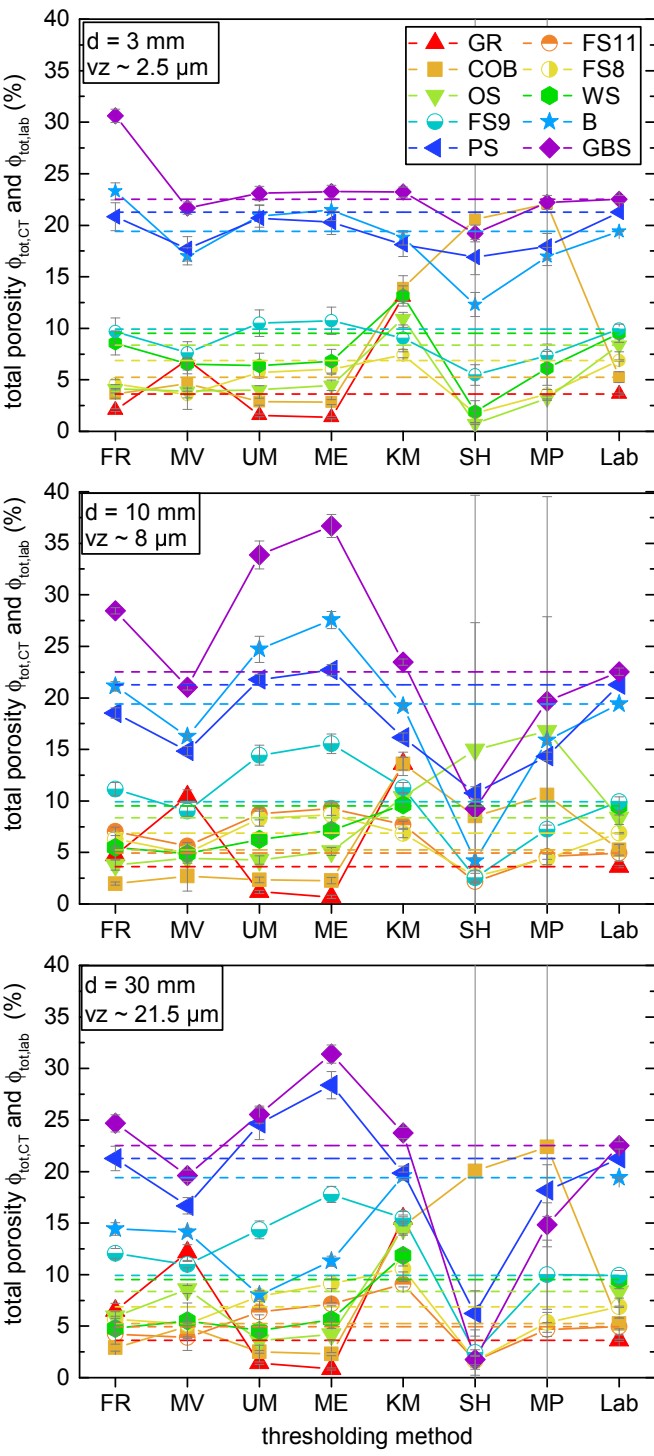

**Figure 9.** Total porosity $\phi_{\text{tot,CT}}$ as a function of sample diameter (d) and corresponding voxel size (vz) calculated from binary images after segmenting with the thresholding algorithms indicated on the horizontal axes. Error bars indicate standard deviations for the nine subvolumes (sample labels as for Figure 8). Dashed lines represent laboratory measurements $\phi_{\text{tot,lab}}$.





In general, the difference between lab and CT porosity increases with increasing sample size and decreasing porosity. Particularly, the total porosities $\phi_{\text{tot,CT}}$ of high porous sandstones such as GBS, PS and B with a diameter of 3 mm are in good agreement with experimentally determined porosities for either FR, MV, UM or ME algorithm. Especially, the MV method yields the overall smallest average deviation from the total porosities measured in the laboratory. Binarization using the SH and MP methods underestimates porosities $\phi_{\text{tot,CT}}$ for the majority of samples, while the KM algorithm tends to overestimate porosity not only regardless of the resolution (Figure 9, COB and GR) but also regardless of the applied denoising filter (Figure 10).

Segmentation of unfiltered images yields generally the best correlation with the porosity determined in the laboratory (Figure 10). Application of denoising filters frequently results in an underestimation of porosity with relative deviations between 15 and 65 % e.g. for the SH and MP algorithm (Figure 10). The smoothing functions implemented in denoising filters such as the Median filter cause this bias due to the removal of minor features. In contrast, images filtered only by NLM result in rather unsystematic porosity values.

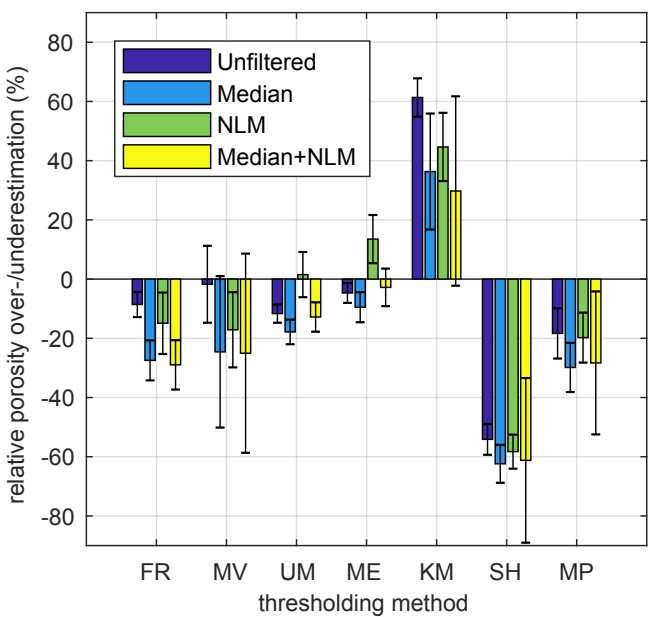

**Figure 10.** Relative deviations $(\phi_{\text{tot,CT}} - \phi_{\text{tot,lab}})(\phi_{\text{tot,lab}})^{-1}$ of calculated CT porosities for the used thresholding methods and applied Median, NLM and Median+NLM denoising filters (sample labels as in Figure 8). The bars represent the average over all subvolumes from all samples, while the error bars indicate the standard deviations.





## 5   Discussion

The link between the shape of the intensity distribution and the resolution observed by us has previously been inferred from numerical binning (e.g., Guan et al., 2018). In image processing, binning refers to a technique for combining a cluster of several pixels to a single pixel with an averaged intensity value. The amount of unresolved pore space incorporated in the

reconstructed gray-value images as partial-volume (PV) voxels increases for decreasing resolutions. By reducing the sample diameter we decreased the voxel size (for a given material) and thereby increased the ratio between pore size and voxel size allowing us to monitor the influence of the partial-volume effect on the intensity distribution. Results for the monomineralic rocks, such as GBS, B or the iron bearing PS, illustrate that a reduction of resolution directly corresponds to an increasing number of PV voxels associated with a transition from bimodality to (left) skewed unimodality (Figure 8). The increasing

number of PV voxels tends to smear out the modes of bimodal distributions associated with the solid grains and pores resulting in skewed, unimodal intensity histograms or a plateau bridging both peaks in the bimodal intensity histograms. These skewed unimodal intensity distributions are provoked by insufficient resolution rather than by micro-structural characteristics of the solid phases in the rocks. Fitting two normal distributions, one for pores and one for the solids, to such bimodal histograms gives a rough estimate of the amount of PV voxels (Figure 11). As evidenced by FS samples, overall porosity plays a subordinate role

for the uncertainty associated with porosity estimates using tomographic images for a given void size distribution; the shape of intensity distributions (Figure 6) remains nearly identical for the porosity values of 9.9, 6.9 and 5.0 % that do not differ in their pore-size distributions (Figure 5). Only the height of the secondary mode associated with open pores is reduced as a function of decreasing porosity within the intensity distribution (see also Lindquist et al., 2000).

Global thresholding methods can only assign one phase to each voxel, which inherently limits the accuracy of these methods

(van Leemput et al., 2003; Kato et al., 2013). Accordingly, the uncertainty for porosity increases with the amount of pore space imaged in PV voxels. In contrast to the tomographic images, our highly resolved SEM images have a maximum resolution of around 20 nm per pixel and enable the identification of features small compared to the X-ray resolution, but at the disadvantage of being two-dimensional and with a limited field of view. Ratios of pore to voxel size close to or even smaller than one result in large measurement errors of about 20 % when porosities are determined by global thresholding algorithms. Porosity estimates

are reliable when the size of a statistically representative quantity of pore-space features exceeds the resolution by at least an order of magnitude (compare Figure 5 with Figure 9). However, the exact limit for the ratio of pore to voxel size varies depending on factors such as total porosity, mineralogical composition, shape of pore space features (e.g. elongated, spherical) etc.

Performing laboratory measurements such as mercury intrusion to determine the amount of pore-throats below the resolution

limit, which can be regarded as an upper limit for unresolved, effective porosity has previously been recommended (Lu et al., 2000; Tanino and Blunt, 2012; Saenger et al., 2016). Mercury intrusion porosimetry measurements can be used as an additional indication for increased uncertainties of CT based porosity estimates. Generally, any comparable laboratory measurements should be performed on the samples being scanned to validate the quantitative porosity estimates, but should not be used to constrain segmentation methods.





Filtering generally introduces a bias due to the removal of minor features resulting in a higher chance of underestimation of porosity. We therefore conclude that filtering generally does not improve porosity determination.

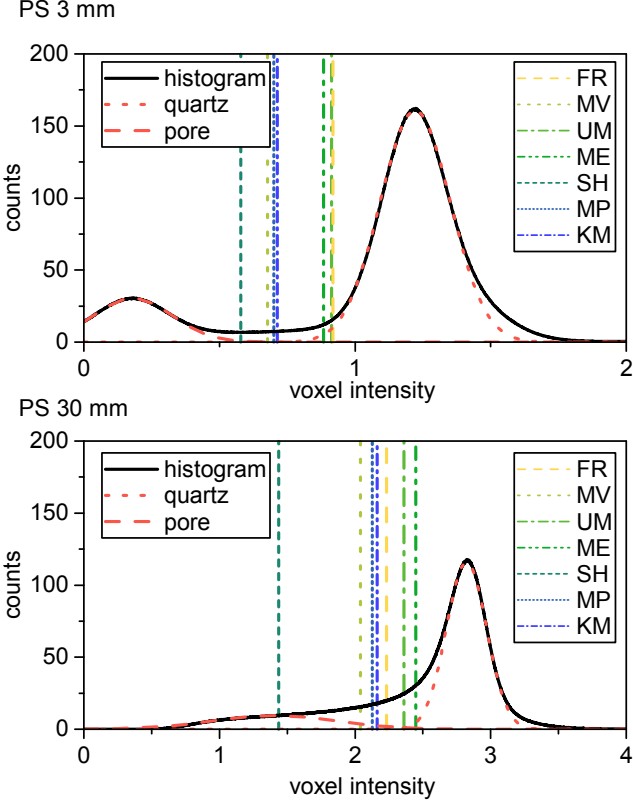

**Figure 11.** Histograms of PS 3 mm and 30 mm diameter illustrate the dependency of the Gaussian distributions on the curvature (class balance) for the measured voxel intensities.

Typical reservoir rocks such as carbonates, shales and breccias were also analyzed in recent studies (Table B2). These rocks are characterized by multimodal pore-size distributions and complex pore geometries. In agreement with our observations,
5 these previous studies found that porosity estimates based on the analysis of tomographic data clearly differ from the laboratory measurements by relative deviations up to more than 50 % (e.g. Fusi and Martinez-Martinez, 2013; Lai et al., 2015; Voorn et al., 2015; Teles et al., 2016). A great majority of the applied evaluation methods for tomographic images tend to underestimate porosity compared to laboratory measurements. Only minor differences between the different lithologies exist. Absolute deviation of porosity is largest for dual energy CT (Teles et al., 2016). Porosity estimates based on tomographic images using
10 synchrotron radiation are slightly better than those based on single energy CT, due to the increased resolution. However, the performance of methods is not only determined by the CT system itself, but is also influenced by the subsequent global and adaptive segmentation methods. The diversity of applied methods hampers the comparison between different studies. Segmen-





tation workflows based on registration and subtraction techniques designated to identify sub-resolution pore space outperform the other methods slightly, but porosity results are still erratic with a high uncertainty (e.g Mayo et al., 2015; Lin et al., 2016).

## 6   Conclusions

We determined the porosity of eight different sandstones using standard laboratory experiments and tomographic images.
We acquired the images with varying resolution and pre-processed the data with denoising filters. Image segmentation was performed using seven different tomography-based thresholding methods. We show by the aid of reconstructed gray-value images and corresponding intensity histograms, that the tomographic pore-volume quantification is unreliable in the light of values gained from conventional laboratory methods. Even for the highest resolution, porosity quantification is imprecise due to (partly) unresolved pore structures, as indicated by analyzing SEM images. Consequently, every digital dataset deduced
from X-ray CT is characterized by rock and resolution specific uncertainties, that are usually greater than the uncertainties of comparable laboratory measurements.

Fairly accurate quantification of porosity from tomographic images can be expected for a ratio of at least 1:10 between voxel size and the size of features of interest, here void size. However, even a relatively high resolution cannot overcome the fundamental shortcomings concerning the determination of porosity from computerized X-ray imaging methods. For example,
polychromatic X-rays, X-ray scattering and system noise contribute and interact with each other to a variable degree in dependence of X-ray energy and material properties. The complex, non-linear relationship of these physical processes hampers an analytical conversion of the measured intensities to density values. The inaccuracy of porosity determination due to technical limitations can be reduced in part by using advanced experimental methodologies such as monochromatic synchrotron radiation or optimized noise reduction methods. These methods are however time-consuming and cost expensive as well as
challenging concerning technical implementation and/or computing power.

In addition to these technical difficulties, general physical characteristics, such as partial-volume effects and sample inhomogeneity, cause errors. It is not possible to distinguish between a non-resolved boundary between two phases or an intermediate material density of a single phase. The uncertainty related to partial-volume effects could only be effectively minimized by increasing the resolution as our data indicates, but the ambiguity of remaining partial volumes remains. All subsequent segmen-
tation and evaluation techniques for determining geophysical rock properties quantitatively are based on certain assumptions about the amount and manifestation of these partial volumes. Generally, the applied global segmentation methods result in a fairly good accuracy for porosity estimates with relative deviations of about 10 % compared to laboratory measurements in case of distinct bimodal intensity distributions. The uncertainty is increased for multiphase systems with overlapping distributions. The shape of the intensity distributions is determined by the overall porosity and its size e.g. macro-, micro- or nano-porosity
and could be used as an indicator for the accuracy of porosity determinations by X-ray computed tomography.

Though quantitative investigations are erroneous when resolution limits are reached, X-ray computed tomography yields unprecedented constraints on the spatial distribution of pore-space features useful for the evaluation of anisotropy, for example. It is recommended to complement the evaluation of tomographic images with (FIB-) SEM analysis to ensure a match between



resolution and structure. Furthermore, X-ray computed tomography offers the opportunity to enhance our understanding of processes such as fluid flow or wave propagation in porous media on the microscale when combined with in-situ experiments.

*Data availability.* The data will be provided by any author upon request



## Appendix A: Sample description

Berea sst (B) is a siliciclastic hydrocarbon reservoir rock with a mean grain size of 250 µm, quarried in Amherst, Ohio (USA). The Berea sst of the Michigan basin is of Early Mississippian age (DeWitt, 1970), terrestrial, fine-grained, well-sorted with closely spaced planar bedding (Churcher et al., 2013). Classified as a subarkose it contains between 6 and 8 % clay minerals,

like kaolinite and illite. The siliciclastic, Crab Orchard sst (COB) from the Cumberland Plateau, Tennessee (USA) is of Pennsylvanian age. The sandstone consists of quartz (approx. 80 to 85 %), feldspar and some lithic fragments. Mica and sericitic clay minerals formed during early diagenesis largely destroyed the primary porosity (Benson et al., 2005). Grains are subhedral to sub-rounded with a mean grain size of approximately 200 µm. The dark brownish sandstone is the only sandstone of our sample suite showing a pronounced fabric and cross bedding on mm-scale.

Fontainebleau sst (FS) is an Oligocene (Rupelian) quartz arenite found in outcrops in the Ile de France region, France (Haddad et al., 2006; French and Worden, 2013). It is fine grained, grain supported, very well sorted and was only buried to shallow depths. The sub-angular grains have grain diameters of about 150 µm. Rarely, kaolinite occurs and secondary amorphous or microcrystalline quartz can be distinguished from primary quartz (French and Worden, 2013). Our suite includes samples with various degrees of siliceous cementation and differing porosities.

Bentheim sst (GBS) is a homogeneous, well sorted, usually yellow colored quartz arenite from the Romberg quarry next to Bad Bentheim (Germany). The average size of the sub-rounded grains is 250 µm. The sandstone was deposited during the lower Cretaceous in a basin north of the London-Brabant Massif and the Rhenish Massif in a shallow marine environment (Kemper, 1976). It consists mainly of quartz (around 90 %), minor K-feldspar or microcline and rare clay minerals. Siliceous cement occurs between quartz grains. Rare opaque minerals occur representing iron-rich minerals, such as hematite or pyrite.

The gray Ruhr sst (GR) is an arkose of upper carboniferous age, deposited in the Variscan foredeep in a fluvio-deltaic environment, and buried and folded during Variscan orogeny (Karg et al., 2005; Jasper et al., 2010). The sandstone occurs in interlayers of silt- and mudstones with a thickness ranging from decimeter to meter scale. The sub-angular to angular grains have an average grain size of about 200 µm. The sandstone consists of around 60 % quartz and approx. 35 % plagioclase heavily altered to sericite. Chlorite as well as illite account for up to 5 %.

Oberster sst (OS) was deposited in central Europe as part of the Keuper sequence, which is composed of sandstones, mudstones, dolostones and evaporites. Deposition took place under hot, semiarid climatic conditions in a fluvio-deltaic and shallow marine environment in the Upper Triassic (Beutler, 2005). The rock is mainly composed of quartz (more than 60 %) and shows microcrystalline pore-filling calcite cement. Other components are feldspars, lithic fragments and dolomite.

Pfaelzer sst (PS) is part of the Lower Triassic series in the German Basin, a large sedimentary basin spread across Poland,
Germany, Denmark as well as the southern realms of the North Sea and Baltic Sea. The sub-arkosic sandstone is characterized by abundant quartz, minor feldspar, rare muscovite and kaolinite as well as accessory zircon. The red color of the samples originates from a minor percentage of hematite. The sandstone is poorly sorted with sub-angular to angular grains of typically 100 to 200 µm in diameter.





Wilkeson sst (WS) is a light gray to brownish, carboniferous sandstone from a quarry in the Wilkeson-Carbonado coal field in the southwest of Washington State, USA (GardJr., 1968). The sandstone is composed of about 50 % quartz and 35 % feldspar highly altered to sericite (Arndt et al., 1980). Minor lithic fragments are black chert and quartzite. The cement is calcitic and/or clayish. Dominant clay minerals are kaolinite and illite. Iron oxides and dolomite occur as well. The rock classifies as a feldspar

5    arenite.




## A1 Thin-section analysis

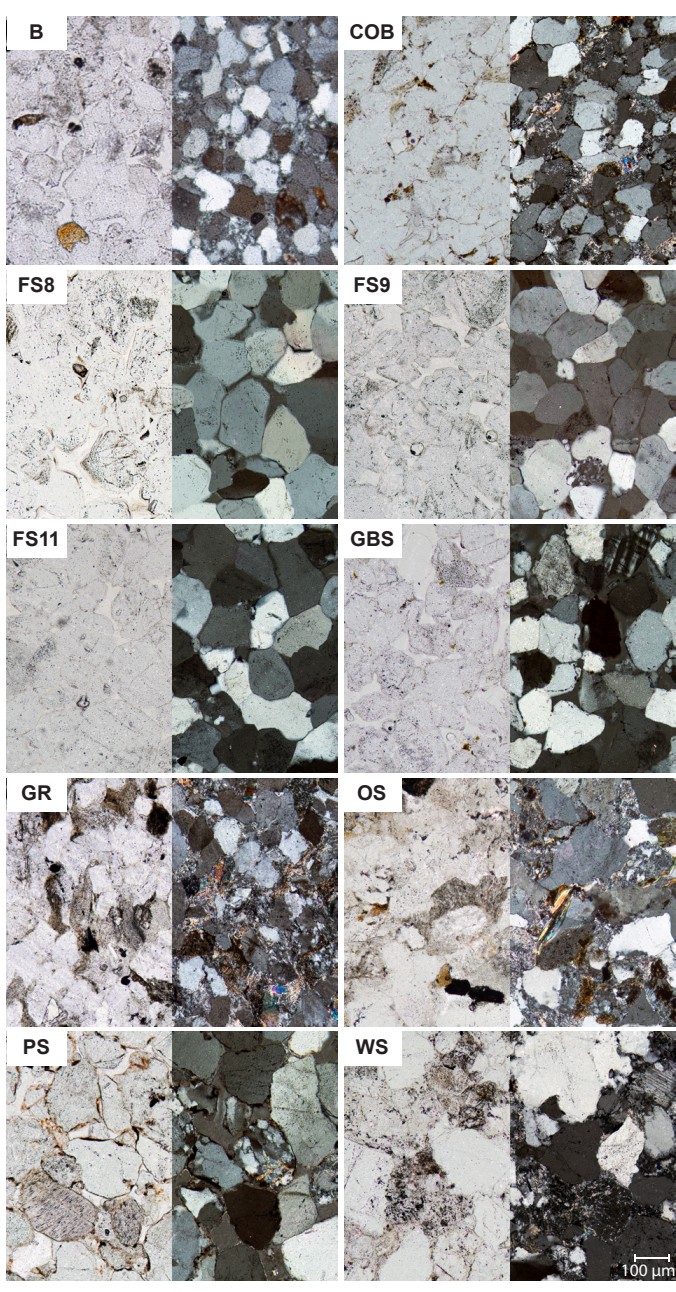

**Figure A1.** Thin sections of (B) Berea, (COB) Crab Orchard, (FS) Fontainebleau, (GBS) Bentheim, (GR) Ruhr, (OS) Oberster, (PS) Pfaelzer(WS) and Wilkeson sandstone under plane (left) and cross-polarized (right) light.





## A2 Error propagation and calibration details

The uncertainty of a function $q$ results from the uncertainties from any number of independent and random measured quantities $x, ..., z$ according to

$$\delta q = \sqrt{\left(\frac{\partial q}{\partial x}\delta x\right)^2 + \cdots + \left(\frac{\partial q}{\partial z}\delta z\right)^2}.$$ (A1)

The uncertainty of the laboratory based measurements of total porosity $\phi_{\text{tot,lab}}$ and effective porosity $\phi_{\text{eff,imb}}$ was calculated using an instrument and machining precision of $0.10 \cdot 10^{-5}$ m for the sample dimensions and a weight uncertainty of $3 \cdot 10^{-7}$ kg. In addition, we assumed an uncertainty of $30$ kg/m$^3$ for fluid density and of $5 \cdot 10^{-10}$ m$^3$ for the pycnometer volume, respectively.

We used argon porosimetry for the determination of effective porosity, which requires an accurate determination of the volumes of the two pressure vessels. Several gas-tight acrylic volumes were used for the calibration procedure. The linear

system of equations for the volumes $V_1$ and $V_2$ can be deduced from relation for pressure after connecting both pressure chambers

$$(p_{1,\text{in}} - p_{2,\text{in}} - p_{\text{final}})V_1 - p_{\text{final}}V_2 = (p_{1,\text{in}} - p_{2,\text{in}} - p_{\text{final}})\sum_{i=1}^{n} V_i - p_{\text{final}}\sum_{j=1}^{n} V_j,$$ (A2)

where $p_{1,\text{in}}$, $p_{2,\text{in}}$ and $p_{\text{final}}$ denote the initial pressure in vessel one, the initial pressure in vessel two and the final pressure of the connected system, respectively. Measurements at various combinations of the standard acrylic volumes placed inside

the vessels were performed to solve the over-determined linear equation system. This method enables the calculation of the dimensionless ratio of pressure changes $P_{\text{K0}}$ according to

$$P_{\text{K0}} = \frac{p_{1,\text{in}} - p_{\text{final}}}{p_{\text{final}} - p_{2,\text{in}}}.$$ (A3)

Plotting $P_{\text{K0}}$ as a function of volume enables the calculation of the slope $m$ and intercept $\Delta y$ (Figure A2), which can be used to calculate the volume of the two pressure vessels

$$V_1 = \frac{1}{m} \quad \text{and} \quad V_2 = V_1 \Delta y.$$ (A4)

The reported uncertainties ()Table 2) account for the uncertainty of length and diameter of the acrylic calibration volumes using an instrument precision of $1 \cdot 10^{-4}$ m. The uncertainty of the pressure transducers is $0.02$ MPa within the measurement range. We determined the standard errors of the vessel volumes by calculating the York fit (York et al., 2004) for the best straight line through our data points. Vessel volumes are $2.3 \pm 0.4 \cdot 10^{-5}$ m$^3$ and $13.4 \pm 4.0 \cdot 10^{-5}$ m$^3$. After calibration of the

two pressure chamber volumes a single measurement of each sample enables the calculation of effective rock volumes $V_{\text{eff}}$ according to (7).





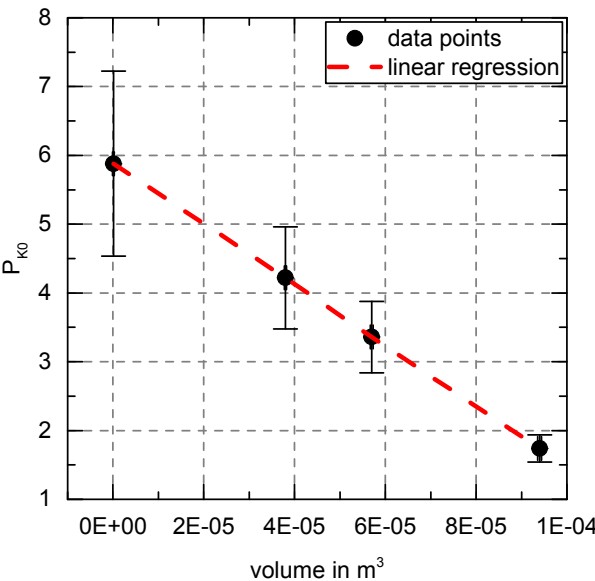

**Figure A2.** Ratio of pressure changes $P_{K0}$ (A3) as a function of the volume of used acryl spacers; slope and intercept of the linear relation enable the calculation of volumes of the pressure vessels (A4).



## Appendix B: Tables

**Table B1.** Experimental setup of X-ray computed micro-tomography (sample labels as in Table 1).

| sample | diameter | SDD[1] | SOD[2] | source mode[3] | acceleration voltage | current | exposure | filter material | voxel-size |
|---|---|---|---|---|---|---|---|---|---|
| | mm | mm | mm | | kV | μA | ms | | μm |
| B | 3 | 490 | 9.5 | MF | 80 | 100 | 100 | 1 mm Al | 2.42 |
| | 10 | 390 | 26 | MF | 80 | 160 | 100 | 1 mm Al | 8.27 |
| | 30 | 465 | 82 | HP | 90 | 150 | 450 | 0.5 mm Cu | 21.87 |
| COB | 3 | 490 | 9.5 | MF | 90 | 50 | 300 | 1 mm Al | 2.42 |
| | 10 | 390 | 27 | MF | 100 | 130 | 100 | 1 mm Al | 8.68 |
| | 30 | 465 | 82 | HP | 90 | 150 | 450 | 0.5 mm Cu | 21.87 |
| FS8 | 3 | 490 | 13 | MF | 90 | 50 | 300 | 1 mm Al | 2.80 |
| | 10 | 390 | 26 | HP | 70 | 180 | 100 | 1 mm Al | 8.34 |
| | 30 | 465 | 82 | HP | 90 | 150 | 450 | 0.5 mm Cu | 21.87 |
| FS9 | 3 | 490 | 13 | NF | 80 | 25 | 1000 | - | 3.30 |
| | 10 | 390 | 26 | HP | 90 | 140 | 100 | 1 mm Al | 8.40 |
| | 30 | 465 | 82 | HP | 90 | 150 | 450 | 0.5 mm Cu | 21.87 |
| FS11 | 10 | 390 | 26 | HP | 90 | 140 | 100 | - | 8.51 |
| | 30 | 465 | 82 | HP | 90 | 150 | 450 | 0.5 mm Cu | 21.87 |
| GBS | 3 | 490 | 12 | MF | 110 | 50 | 100 | 1 mm Al | 3.04 |
| | 10 | 390 | 26 | HP | 80 | 160 | 100 | 1 mm Al | 8.42 |
| | 30 | 465 | 82 | HP | 90 | 150 | 450 | 0.5 mm Cu | 21.87 |
| GR | 3 | 490 | 12.5 | NF | 80 | 25 | 1000 | - | 3.12 |
| | 10 | 390 | 26 | HP | 100 | 100 | 100 | 1 mm Al | 8.23 |
| | 30 | 465 | 82 | HP | 90 | 150 | 450 | 0.5 mm Cu | 21.87 |
| OS | 3 | 490 | 9.5 | MF | 80 | 90 | 100 | - | 2.42 |
| | 10 | 390 | 26 | HP | 90 | 120 | 100 | 1 mm Al | 8.21 |
| | 30 | 465 | 82 | HP | 90 | 150 | 450 | 0.5 mm Cu | 21.87 |
| PS | 3 | 490 | 10 | MF | 100 | 50 | 300 | 1 mm Al | 2.64 |
| | 10 | 390 | 26 | HP | 90 | 130 | 100 | 1 mm Al | 8.21 |
| | 30 | 465 | 82 | HP | 90 | 150 | 450 | 0.5 mm Cu | 21.87 |
| WS | 3 | 490 | 13 | MF | 90 | 50 | 300 | 1 mm Al | 2,80 |
| | 10 | 390 | 26 | HP | 90 | 120 | 100 | 1 mm Al | 8.35 |
| | 30 | 465 | 82 | HP | 90 | 150 | 450 | 0.5 mm Cu | 21.87 |

[1]SDD denotes the source-detector distance and [1]SOD denotes the source-object distance. [3]Source mode abbreviations NF (nanofocus), MF (microfocus) and HP (high power) denote the focal spot diameters of 1 μm, 5 μm and 25 μm, respectively.





**Table B2.** Literature review of porosity determination for different rock types, which are highlighted by symbols, while superscript numbers indicate different laboratory methods and superscript letters indicate the different X-ray computed tomography methods.

| rock type | sample name | author | porosity lab 1 | porosity lab 2 | porosity CT method 1 | segmentation method 1 | porosity CT method 2 | segmentation method 2 | voxel-size |
|---|---|---|---|---|---|---|---|---|---|
| | | | % | % | % | | % | | μm |
| carbonate | Oolite | Mayo et al. (2015) | 29.0[1] | 38.0[2] | ■ 29.2 | R&S | | | 3.2 |
| | Oolite | | 27.0[2] | | ■ 28.8 | R&S | | | 3.2 |
| | Darai | | 16.0[1] | | ■ 13.9 | R&S | | | 3.2 |
| | PGI | | 2.0[1] | 3.0[2] | ■ 2.0 | R&S | | | 3.2 |
| | Grosmont | Andrä et al. (2013) | 21.0[1] | | ▲ 23.3±3.8 | MBW,MV,OM | | | 2.0 |
| dolomite | Indiana | Teles et al. (2016) | 16.9[2] | | ▲ 4.7±0.6 | MBW | ♦ 7.3 | MBW | 25.0 |
| | Pink | Freire-Gormaly et al. (2015) | 29.0[5] | 34.0±3.0[4] | ▲ 30.0±2.0 | MV | ▲ 35.0±2.0 | MV | 10.0 |
| | Amabel | Hussein et al. (2015) | 12.8±1.9[1] | | ● 9.1 | R&S | | | 18.2 |
| fractured dolomite e.g. breccia | Ambarino | Fusi and Martinez-Martinez (2013) | 4.5[3] | | ▲ 1.1 | manual | | | 24.0 |
| | Beige Serpiente | | 2.1[3] | | ▲ 1.2 | manual | | | 24.0 |
| | - | Voorn et al. (2015) | 9.7[2] | | ▲ 3.0 | MSHF | | | 12.5 |
| | - | | 3.8[1] | | ▲ 0.9 | MSHF | | | 12.5 |
| | - | | 5.1[2] | | ▲ 4.2 | MSHF | | | 12.5 |
| | - | | 5.1[2] | | ▲ 1.8 | MSHF | | | 12.5 |
| | - | | 5.8[1] | | ▲ 0.9 | MSHF | | | 12.5 |
| | - | | 8.9[2] | | ▲ 2.6 | MSHF | | | 12.5 |
| | - | | 6.7[1] | | ▲ 2.0 | MSHF | | | 12.5 |
| limestone | Indiana | Teles et al. (2016) | 13.8[2] | | ▲ 1.4±0.2 | MBW | ♦ 4.92 | MBW | 25.0 |
| | Portland | Lin et al. (2016) | 19.5±0.6[2] | | ● 18.9 | R&S+MV | ● 19.3 | R&S+MBW | 5.0 |
| | Estaillades | | 29.3±0.7[2] | | ● 29.8 | R&S+MV | ● 28.8 | R&S+MBW | 5.0 |
| | Oolithe Bl. | Boone et al. (2014) | 13.8±0.3[5] | 10.6[3] | ▲ 1.45 | MV | ● 5.75 | R&S | 4.5 |
| | Oolithe Bl. | | 10.7±2.5[5] | 11.6[3] | ▲ 7.19 | MV | ● 15.7 | R&S | 4.5 |
| | Oolithe Bl. | | 2.1[3] | | ▲ 2.0 | MV | ● 4.01 | R&S | 4.5 |
| | Crema Valencia | Fusi and Martinez-Martinez (2013) | 0.1[3] | | ▲ 1.0 | manual | | | 24.0 |
| | Rojo Cehegin | | 0.4[3] | | ▲ 3.7 | manual | | | 24.0 |
| | Blanco Alconera | | 0.2[3] | | ▲ 2.4 | manual | | | 24.0 |
| | Travertino Bl. | | 6.6[3] | | ▲ 2.9 | manual | | | 24.0 |
| | Indiana | Freire-Gormaly et al. (2015) | 19.0[5] | 23.0±4.0[4] | ▲ 13.0±1.0 | MV | ▲ 14.0±5.0 | MV | 10.0 |
| | Edwards | Lai et al. (2015) | 45.0[2] | | ▲ ca. 38.0 | MBW | | | 1.0 |
| | Guelph | | 9.0[2] | | ▲ ca. 4.0 | MBW | | | 1.0 |
| | Estaillades | | 28[2] | | ▲ ca. 17.0 | MBW | | | 1.0 |
| | Ketton | | 23.0[2] | | ▲ ca. 11.0 | MBW | | | 1.0 |
| | Indiana | | 18.0[2] | | ▲ ca. 12.0 | MBW | | | 1.0 |
| marble | Amarillo Triana | Fusi and Martinez-Martinez (2013) | 4.9[3] | | ▲ 0.1 | manual | | | 24.0 |
| | Blanco Tranco | | 0.2[3] | | ▲ 1.0 | manual | | | 24.0 |
| | Gris Macael | | 0.3[3] | | ▲ 6.4 | manual | | | 24.0 |

**Lab methods:** 1 = gravimetric, 2 = HE injection, 3 = mercury intrusion porosimetry (MIP), 4 = SEM, 5 = unknown

**CT methods:** ▲ = single energy, ■ = synchroton, ♦ = dual energy, ● = contrast agent

**Segmentation:** MV = maximum variance algorithm, MP = Mid-Point thresholding, R&S = registration and substraction of images, MBW = marker-based watershed, MSHF = multiscale Hessian filter, HYST = hysteresis thresholding, IC = indicator kriging, AC = active contour algorithm, OM = other methods;



**Table B3.** Continued Table B2.

| rock type | sample name | author | porosity lab 1 % | porosity lab 2 % | porosity CT method 1 % | segmentation method 1 | porosity CT method 2 % | segmentation method 2 | voxel-size μm |
|---|---|---|---|---|---|---|---|---|---|
| sandstone | Idaho gray | Teles et al. (2016) | 28.2[2] | | ▲ 17.3±0.7 | MBW | ♦ 21.5 | MBW | 25.0 |
| | Castlegate | Mayo et al. (2015) | 28.0[1] | 25.0[2] | ■ 22.4 | R&S | | | 3.2 |
| | Bray | Cnudde et al. (2011) | 14.0[1] | | ▲ 18.0 | OM | | | 7.4 |
| | Fontainebleau | Andrä et al. (2013) | 15.2[5] | | ■ 14.7 | MBW,MV,OM | | | 7.5 |
| | Berea | | 20.0[2] | | ■ 19.7±1.3 | MBW,MV,OM | | | 0.7 |
| | Berea | Lai et al. (2015) | 21.0[2] | | ▲ ca. 13.0 | MBW | | | 1.0 |
| | Berea | | 23.0[2] | | ▲ ca. 15.0 | MBW | | | 1.0 |
| | Berea | Leu et al. (2014) | 19.9[5] | | ■ 16.6 | MBW | ■ 18.7 | HYST | 2.9 |
| | Berea | Peng et al. (2012) | 22.8[2] | 23.2[3] | ▲ 15.2 | MP | | | 12.7 |
| | Berea | | 22.8[2] | 23.2[3] | ■ 19.4 | MP | | | 2.9 |
| | Berea | Prodanović et al. (2007) | 22[1] | | ■ 18 | IC | | | 4.9 |
| | Shelf sandstone | Bhattad et al. (2014) | 25.2[2] | | ■ 22.6 | R&S+AC | | | 15.0 |
| | Shelf sandstone | | 32.8[2] | | ■ 31.8 | R&S+AC | | | 15.0 |
| | Chats-worth | Hussein et al. (2015) | 12.1±0.8[1] | | ● 13.1 | R&S | | | 18.2 |
| | Westphalian | van Geet et al. (2003) | 10.5[1] | 10.5[3] | ▲ 11.4 | MV | | | 22.0 |
| | Westphalian | | 11.1[1] | 11.0[3] | ▲ 12.1 | MV | | | 22.0 |
| shale | | Ramandi et al. (2016) | 8.0[1] | 12.8[2] | ● 9.4 | R&S+AC | | | 5.5 |
| | Muderong | Mayo et al. (2015) | 47.0[1] | 14.0[2] | ■ 23.4 | R&S | | | 3.2 |

**Lab methods:** 1 = gravimetric, 2 = HE injection, 3 = mercury intrusion porosimetry (MIP), 4 = SEM, 5 = unknown

**CT methods:** ▲ = single energy, ■ = synchroton, ♦ = dual energy, ● = contrast agent

**Segmentation:** MV = maximum variance algorithm, MP = Mid-Point thresholding, R&S = registration and substraction of images, MBW = marker-based watershed, MSHF = multiscale Hessian filter, HYST = hysteresis thresholding, IC = indicator kriging, AC = active contour algorithm, OM = other methods;

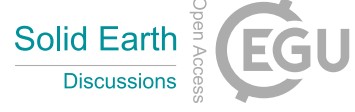

*Author contributions.* TEXT

*Competing interests.* The authors declare that they have no competing interests.

*Disclaimer.* TEXT

*Acknowledgements.* The first author benefited from a PhD grant from the graduate school "Applied Research on Enhanced Geothermal
5   Energy System (AGES)" (AZ: 321-8.03.04.03-2012/x), a joint venture between the International Geothermal Centre Bochum, Bochum
University of Applied Sciences, and the Institute of Geology, Mineralogy and Geophysics, Ruhr-Universität Bochum.





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
