# Peer review of "Evaluating porosity estimates for sandstones based on X-ray micro-tomographic images"

_Solid Earth, 2019_

## Referee Comment (RC1) · Anonymous Referee #1 · 12 Mar 2019

The work focuses on determining the impact of micro-CT imaging and image processing on the porosity of rock samples. This topic remains a topic of interest, because improvement in X-ray technology extend the field-of-view / resolution operating window and improvements in image processing also provide potentially more robust and noise/artifact tolerant algorithms. The issue on the resolution limit has indeed been reported in the Leu et al. 2014 reference but potentially already earlier (see references in the Leu et al. 2014 paper). There are several follow-up studies, i.e. the one by C Soulaine, F Gjetvaj, C Garing, S Roman, A Russian, P Gouze . . . Transport in porous media 113 (1), 227-243, 2016. There are also several more recent follow-up studies that build on the simulated Hg-air instrusion workflow introduced in the Leu et al. pa-

per and investigate the impact of resolution limit on porosity and permeability much more systematically, N Saxena, R Hofmann, F O. Alpak, J Dietderich, S Hunter, R J. Day-Stirrat, Effect of image segmentation & voxel size on micro-CT computed effective transport & elastic properties, Marine and Petroleum Geology, 2017. N Saxena, A Hows, R Hofmann, FO Alpak, J Freeman, S Hunter, M Appel, Imaging and computational considerations for image computed permeability: Operating envelope of Digital Rock Physics, Advances in Water Resources 116, 127-144, 2018.

First, It would be good to reference this work as it is very fundamental in nature and provides a conceptual limit to which extent the porosity in a certain rock is accessible by imaging at a given resolution. The work by Saxena et al. suggests that for porosities below 20% there are in many cases systematic differences between imaged and independently measured porosity, but it still depends on the actual pore size distribution. I would therefore encourage the authors to apply the workflow described in the Saxena papers and estimate for each situation which fraction of the porosity is not accessible because of the resolution limit. The second aspect that is important to mention is that porosity alone is perhaps not a unique enough parameter to determine the validity of a segmentation. It is much more important to consider porosity, and also permeability. The findings of Saxena et al. is that the resolution limit leads to a systematic under-prediction of porosity and/or overprediction of permeability, depending on how thresholds are chosen. But for an image below the resolution limit it is not possible to match both.

When it comes to segmentation algorithms, the focus is on absolute thresholding methods. It would be interesting to consider also other thresholding methods such as Watershed, C-means clustering, indicator Kriging, and perhaps also machine learning based methods like the trainable WEKA segmentation that is available in the OpenSource FIJI package. A recent overview is presented in S Berg, N Saxena, M Shaik, C Pradhan, Generation of ground truth images to validate micro-CT image-processing pipelines, The Leading Edge 37 (6), 412-420, 2018.

Concerning the imaging hardware, it would be important to determine the physical resolution of the system. I am not sure if the system is capable of achieving 2.5 micrometer physical resolution. The authors should verify that by using a JIMA test pattern for the relevant micro-CT settings.

Looking at Fig. 7, from my own experience, the NLM filter should perform better. I am not sure why the center of grains remains largely unfiltered. It could be a setting issue or input image quality issue. The 3D NLM filter available in GeoDICT or Avizo should perform better. Furthermore I would never consider using a median filter because of its impact on the sharpness on grain boundaries and the associated resolution limit issue. In the Leading Edge paper mentioned above, the best results are theoretically obtained without filtering, using a segmentation method that is robust against filtering.

---

## Referee Comment (RC2) · Edward Andò (Referee) · 29 Mar 2019

I think this paper presents a careful study of the measurement of porosity in sandstones – a measurement of primary importance for the mechanical behaviour and permeability. The paper is essentially what I wold consider a technical note, so most of my comments are technical. The paper is interesting to publish but it has some flaws. In order of importance:

1. The take away message is quite specific and quite difficult to digest. This is, in my opinion related to the lack of theoretical results regarding porosity measure-mement which could both render conclusions more general, but also make the

paper more appealing. An example of what I'm saying is to study the effect of signal-to-noise, blurring and pore size on the error of porosity estimation on a synthetic test case with only two phases. This could easily be done by clipping correctly-calculated partial volume spheres representing pores out of a block of material (what mathematical morphologists call a meatball geometry). This will allow some expectations of how these three key parameters affect the measurement, allowing your real measurements to be put into context, and yielding useful relations about blur vs pore size for accurate porosity measurements for example. Furthermore filtering can be tested synthetically.

2. Some strange choices have been made on treatment of the x-ray tomography volumes. First of all there are some clear artefacts in the high-resolution images that are not really commented. I'm afraid that Median and NLM are really odd choices – median filtering is really only used by professional for removing extreme values, and NLM is so non-linear that it's behaviour is difficult to characterise. It's surprising that grey-mass preserving filter such as averaging (and without thinking about it too hard Gaussian blurring should also be) would be more suitable. For more advanced filtering Bilateral filtering as discussed abundantly in the literature from the Canberra group is a much better choice since it is local. A number of parameters are missing (structuring element for filters?). Another key point is that a very strange choice has been made to study thresholded images. If you believe that the partial volume effect holds, and that greylevels are a linear function of voxel occupancy (reasonable) for a two-phase material you can calibrate the greylevel of pores and solids and assign a porosity to each voxel with an interpolation. This calibration will remove the thresholding step and should return much less noisy images. Again this can be proved on a synthetic example as suggested above.

3. A number of laboratory measurements have been made. This is really good to compare to the x-ray measurements, however the different measurements do not seems to be consistent between each other (different non-tomography techniques do not fall into the error band of each other). This key point needs to be analysed and discussed further... different ways of measuring the same thing must yield values within error, otherwise the same thing is not being measured (or errors are underestimated).

Furthermore, I am not exactly in this field, so perhaps habit are different, but I find in a number of places that overly general and not supported by facts that are presented. Some of the language surrounding x-ray acquisition and reconsrtuction is a little bit inexact, and I have offered some comments to improve this.

Comment before starting to read this paper: judging from the layout in the abtract, I would like to make the following predictions, that we could take as a reasonably expert's guess:

- Otsu is best

- No filtering is best (unless gaussian blur is used). To quote J.C. Santamarina "The best amount of filtering is none"

- pores smaller than 5*pixel size are a problem

Synchrotrons do not generally produce monochromatic beams (wigglers and bending magnets don't in any case...), but generate enough flux that the beam can be monochromatised while keeping good performance.

There is normally a negative sign in the exponent, unless your $\mu$ values are negative. It may be worth talking about the dependence of $\mu$ on x-ray energy, which will allow beam hardening to be introduced with more ease later.

Artifacts? Artefacts.

"This non-uniqueness".

What is non unique? I propose to say "this spatial averaging process"

"...results in a blurring of all phase boundaries in a CT image."

This is not blurring, it is the manifestation of the discretisation of the measurement. I agree that boundaries of phases will be filled with partial volume voxels, but this is an expected result of voxel-volume averaging.

This sentence is not homogeneous:

"The spatial resolution of the tomographic image produced by conventional X-ray CT is limited by the diameter of the X-ray source, the X-ray photon energy, the absorption of the sample as well as the number and size of pixels on the detector"

"However, the (isotropic) voxel size is the calculated size of a discrete cubic volume element in the reconstructed image, which can actually be lower than the actual spatial resolution of the image"

Sure, but it's a pity to cop out of measuring at least the radiograph resolution with standard techniques.

In my experience, off focal radiation is a detail compared to mechanical stability of imaging system. Maybe also with noting phase contrast effects?

"Standard CT systems do not record voxel intensities in terms of any physical unit but rather map the data to a somewhat arbitrary range meeting either the data storage requirements (e.g. 0 to 65535 for common 16-bit integer dataformats) or, to ensure compatibility of images from different CT systems, transforming the data to so-called Hounsfield units or CT numbers." ...and later... "Due to the non-physical units of the image data"

Weeell, no, I don't really agree. Non-physical units means that they are units what make no physical sense (to me). Here the units of the reconstructed image are a complex energy-integrated attenuation coefficient which you cannot express the units for, but this does not mean they do not exist. 16-bit integers are only used as output formats, all reconstruction codes are at least float internally for numerical reasons, so "record voxel intensities" is also wrong. Furthermore, a detector records, here we are reconstructing, computing, calculating, inverting but not recording.

Aesthetic comment: please don't abbreviate sandstone to sst, why? to save 5 characters? It decreases readability for no reason.

Rabbani is an odd reference for the watershed...

Reconstruction process is back projection? "Convolution and back filter" is really not clear.

> "...mapped to a 16-bit integer range between zero, corresponding to the background, in our case air, and 65535, corresponding to the detection limits of the CT-Scanner..."

For quantitative studies (this like one is aiming to be) cutting the reconstructed values at zero (zero what?) is bad practice, since you miss the noise distribution below zero for the background. Re-reading the sentence, I'm afraid to say that it's all meangingless and should be re-written – zero is not defined, and the "detection limits of the CT-Scanner" are not well defined (with a lump of lead what are your detection limits?).

After citing so much work by Sheppard I'm sad to see that you are not going to try a Bilateral filter which is a clear and well-defined local filter compated to the complex and hard-to-characterise NLM filter.

Details of NLM filter setting please.

GBS points counting way off, probably copy-paste error with below, PLEASE CHECK!

Comment on the fact that all measurements of porosity in Table 2 with their associated errors rarely fall in the same margins! This is a clear indication that the errors are extremely underestimated, or that these porosity measurements are in fact measuring different things. I'm sorry if this is a trivial point, but it's still very surprising. I would really like an answer to this point.

"Images obtained using X-rays pre-hardened by filters have a higher signal to noise ratio than those taken without a filter..."

Why such a general sentence so far in the document? This is not true in many cases, like in biological samples.

"absolute voxel intensities are displayed using a uniform gray value range"

This is a bit rich considering you've said that they are non-physical... See comment above.

"The quality of images taken with a focal spot diameter of 1 $\mu$m and without X-ray filters (Figure 6, FS9) is similar to images recorded with a focal spot diameter of 5 $\mu$m using aluminum filters (Figure 6, FS8)."

This is nonsense, remove it. Blur and contrast are two independent axes of image quality – which may be coupled in your system, sure, but you really should not feel empowered to make this kind of general statement, it's utterly undefendable.

Figure 6, there is a fair amount of geometrical inaccuracy in the reconstructions with small pixel sizes, which looks like motion blur, which should be mentioned (FS9, WS, OS, FS8), could this be a cause of skewness?

Non-local means example (c, g) in Figure 7 does not fill me with joy, it's a very strange result of filter, with some edges sharpened and some texture remaining in the grains. This is the fundamental problem with NLM, that filtering is not homogeneous or isotropic, and there is little control of how and where the algorithm chooses to filter. I've extracted the original image from the PDF, and run 20 iterations of Bilateral, which denoises nicely – if I can upload them with my review I will. The blurred edges due to motion blur seem to be interpreted in a strange way by the filter.

Figure 8 shows a very dramatic change of the automatic thresholds. In some cases the second peak is very hard to see – adding the log of the pixel count (which is quite typical) would help in this regard.

> "In general, the difference between lab and CT porosity increases with increasing sample size and decreasing porosity."

This is pretty much expected since you are using a thresholding approach. On page 15 line 10, I think a more theoretical discussion about grey-scale-preservation in filtering (not guaranteed by either of the filters used, unlike an average filter) will help make the discussion more precise.

> "We therefore conclude that filtering generally does not improve porosity determination."

Woah there, the choice of two strange filters does not allow such a general conclusion!!
* * *
[Figure]

**Fig. 1.**

[Figure]

**Fig. 2.**

---

## Referee Comment (RC3) · Anonymous Referee #3 · 2 Apr 2019

The work focuses on the effects of image resolution, segmentation and filtering methods on the evaluation of bulk porosity in different porous sandstones through CT imaging. I think the paper represents a good contribution to scientific progress, as it outlines the limitations of x-ray microtomography in evaluating bulk porosity. However, I think their data analysis and discussion could be improved substantially. In particular, the choice of the applied methods can be debated: there are many more robust segmentation algorithms and filtering methods that would perform better and will therefore have a different impact on the final porosity quantification. The choice of the authors for these methods have not been explained, hence their final conclusions are weak.

[Figure]

Overall, the presentation of the data is well structured. Some figures could be improved (see specific comments). Some sentences can be structured better – they are hard to read for a non-expert in the XCT field (see specific comments).

I outlined below the major comments I have, and some specific comments later on.

Some general comments

1. First at all, the true spatial resolution (which is different from the voxel size) may depend on the sample type (e.g. grain size, composition), sample size, X-ray spot size and the precision of the alignment of the source, rotation axis and camera as well as the geometric magnification given by the ratio of distances between the source, sample and camera. Check Feser et al. (2008). In Line 9 -16, I think you should clarify better the difference between voxel size and true spatial resolution.

Diameter of the x-ray source, do you mean the x-ray spot size? Please be consistent in terminology.

2. The choice of the segmentation methods is a bit strange. If the aim of the paper was to assess the effects of different segmentation methods, you cannot base your findings only on global thresholding methods.

Line 34 – 35: Segmentation is the process to identify and separate phases, but it is not just based on threshold values. There are many algorithms (Indicator Kriging, C-means clustering, machine-learning algorithms such as the Trainable Weka, or watershed) that takes into account other parameters, in addition to the intensity. This misleads the reader, as the text here implies that segmentation is only possible though global thresholding, which is not true. In fact, paragraph in Line 5 should be tied up better to previous comment. In other words, why did you choose only global thresholding algorithms? Your choice is not explained.

3. I also disagree on the choice of the filtering methods. First at all, the median filter heavily smooths the edges of the objects, and the image becomes less sharp,

impacting in the resolution. In addition, a kernel size of 3 is quite large – what does it mean that you applied the filter in 3 different successions? Is that referring to 3 different kernel sizes? And what about the impact of different kernel size or different neighbouring (6 vs 26)? The authors should consider that also these parameters may have an impact in the final quantification (and they usually do). In Figure 7, I would expect the NLM filter to perform better – what about the influence of the choice of parameters for the filter? Did the authors verify this?

There are many other denoising filters, that are definitely a better choice than the median filter, such as bilateral filter or Anisotropic Diffusion. In addition, the results of the filtering will significantly depend also on the choice of the parameters i.e. number of iterations.

4. Something minor to consider, but since you extrapolated subvolumes, would pores lying on the edges have an effect on the final quantifications? It would be interesting to repeat the analyses, by using "exclude edges" or "border kill" in Avizo. Also, I would add error bars in any quantification: since you are trying to assess the effect of different segmentation methods, when applying these methods your choice of the threshold is also subjected to error, therefore error bars are needed in order to estimate these errors. By using morphological operators such as erosion and dilation. See Fusseis et al (2012) for example.

Page 27, line 3 "We therefore conclude that filtering generally does not improve porosity determination" – Well, as mentioned in the points above, such a statement is odd considering that only 2 filters (and a mix of the 2) were chosen, while some more robust denoising filters are out there and should be included in this analysis for a more coherent discussion.

Specific comments:

1. Avoid sentences like "as one would" or "as one distinguishes" – always better to use a third person narrative style.

2. Line 15: "Owing to its relevance"?

3. Page 3 line 3: reconstruction artifacts? I think it is artefacts.

4. You should explain what a voxel is, when introducing this term for the first time. A reader non-expert in the field would not know what a voxel is.

5. Page 4 line 18: what is sst? It took me a while to understand what it means, and please avoid using acronyms that i) are not scientific ii) you did not explain earlier on.

6. Line 19 page 4: how did you check for this? what if anisotropies were present at the microscale?

7. Is 6.6. mm2 a representative area?

8. Assuming the samples are isotropic: what if they are not? In other words, it should have been tested.

9. Page 7 line 11-13: how did you account for errors derived from these measurements? Or are they extremely precise? Not an expert on the matter.

10. Page 8 line 21: "Each pixel in the detector has an edge length of 127 $\mu$m and a gray-value resolution of 14 bit resulting in a nominal system resolution of around 1 $\mu$m" This sentence originally confused me – is it really necessary? I think it would be better to keep either nominal resolution or true spatial resolution, but not both as the reader may get confused.

11. Page 8 line 26: How is the resolution controlled by the magnification? Get equation (see Feser et al. 2008). Also, why is this mentioned here and not at the beginning, with the rest? Be consistent.

12. Page 9 line 20: I would put in brackets the real volume dimensions either in mm3 or um3.

13. Page 13, lines 21-25: this sentence is an interpretation and should be part of the

discussions.

14. Page 14 I forgot what B, GBS and PS indicate. I would probably put the label next to them. Also, later in line 10 – you first describe the sandstones by using their relative content in porosity (highly porous sandstones) but then you call Group C, and I found this confusing: either you use the group names (since you just defined them) or the porosity content, just be consistent and do not mix between one definition and the other when listing them.

15. Line 13: the mean diameter falls below the resolution – does that mean that there is a lot of porosity missing from the CT analyses? And 211 designated pores – are these derived from SEM? As I am getting confused.

16. Figure 4, error bars: have these been defined in the text? I did not find it, maybe I missed it.

17. Figure 5: The eq. diameter here is calculated from SEM images, so it is a 2D calculation. At the same time, you mentioned the voxel size of the XCT and make a comparison between the voxel size and the mean pore diameter. However, the mean pore diameter is a 2D measurement and therefore this comparison in my opinion does not make much sense: it would be better to use the eq diameter derived from XCT images or make a comparison of the eq diameters derived from SEM and XCT. In that sense, porosity is not the only parameter to determine, but also pore throats, pore connectivity and permeability.

18. Table 2: P-wave velocity – existing measurements from where?? Or did you calculate them?

19. Figure 6: I think it should include the type of filter used, otherwise the reader has to go back and forth to table B1. A figure should be self-contained.

20. Figure 8: Where is GBS? In line 27 page 20 you refer to GBS in figure 8 I think.

21. Page 28 Lines 12 – 19: For very heterogeneous samples, scientists normally

adopt a multi-scale imaging approach to determine realistic values of porosity and permeability, overcoming the issue of low resolution or not representative samples. Something to keep in mind for the discussion perhaps.
* * *

---

## Author Comment (AC1) · 4 May 2019

Please find our answers to the comments in the Supplement.

Please also note the supplement to this comment:
https://www.solid-earth-discuss.net/se-2019-48/se-2019-48-AC1-supplement.pdf
* * *

---

## Author Comment (AC2) · 4 May 2019

The work focuses on determining the impact of micro-CT imaging and image processing on the porosity of rock samples. This topic remains a topic of interest, because improvement in X-ray technology extend the field-of-view / resolution operating window and improvements in image processing also provide potentially more robust and noise/artifact tolerant algorithms.

We appreciate this general evaluation of the reviewer.

1.) The issue on the resolution limit has indeed been reported in the Leu et al. 2014 reference but potentially already earlier (see references in the Leu et al. 2014 paper). There are several follow-up studies, i.e. the one by Soulaine, F Gjetvaj, C Garing, S Roman, A Russian, P Gouze . . . Transport in porous media 113 (1), 227-243, 2016. There are also several more recent follow-up studies that build on the simulated Hg-air instrusion workflow introduced in the Leu et al. paper and investigate the impact of resolution limit on porosity and permeability much more systematically, N Saxena, R Hofmann, F O. Alpak, J Dietderich, S Hunter, R J. Day-Stirrat, Effect of image segmentation & voxel size on micro-CT computed effective transport & elastic properties, Marine and Petroleum Geology, 2017. N Saxena, A Hows, R Hofmann, FO Alpak, J Freeman, S Hunter, M Appel, Imaging and computational considerations for image computed permeability: Operating envelope of Digital Rock Physics, Advances in Water Resources 116, 127-144, 2018.

We are grateful for the hints toward this body of important previous work and will appropriately refer to it in the revised manuscript.

2.) The work by Saxena et al. suggests that for porosities below 20% there are in many cases systematic differences between imaged and independently measured porosity, but it still depends on the actual pore size distribution. I would therefore encourage the authors to apply the workflow described in the Saxena papers and estimate for each situation which fraction of the porosity is not accessible because of the resolution limit.

We fully agree with the reviewer's comment on the crucial role of pore-size distribution. We will expand our discussion of the role of resolution by accounting for Saxena at al.'s work.

However, we are not aware of a workflow by Saxena which can be used to determine the porosity more accurately. The basic problem is as discussed in this paper is the resolution. The calibration ideas by Saxena mostly improve the estimation of effective transport and elastic properties in comparison to laboratory results.

3.) The second aspect that is important to mention is that porosity alone is perhaps not a unique enough parameter to determine the validity of a segmentation. It is much more important to consider porosity, and also permeability. The findings of Saxena et al. is that the resolution limit leads to a systematic under-prediction of porosity and/or overprediction of permeability, depending on how thresholds are chosen.

We agree with the reviewer regarding the crucial importance of permeability. Here, we focused on porosity because it is a) the fundamental volumetric microstructural parameter for all transport properties, i.e., electrical conductivity and wave velocity in addition to hydraulic conductivity, and b) it is a fairly direct outcome of the tomographic analysis and does not require further calculations (e.g., lattice Boltzman procedures) with their own issues. We will clarify our objective and perspective regarding this point and will dwell on consequences of our findings for transport properties referring, e.g., to the work of Saxena et al. mentioned by the reviewer and also Soulaine, 2016 (DOI: 10.1007/s11242-016-0690-2).

4.) When it comes to segmentation algorithms, the focus is on absolute thresholding methods. It would be interesting to consider also other thresholding methods such as Watershed, C-means clustering, indicator Kriging, and perhaps also machine learning based methods like the trainable WEKA segmentation that is available in the OpenSource FIJI package. A recent overview is presented in S Berg, N Saxena, M Shaik, C Pradhan, Generation of ground truth images to validate micro-CT image-processing pipelines, The Leading Edge 37 (6), 412-420, 2018.

We deliberately focused on global thresholding methods to show their reliability/limits for the determination of porosity. Global thresholding methods are computational inexpensive and simple to implement making them ideal candidates for unsupervised segmentation of large datasets of reconstructed intensity images without prior knowledge. Furthermore, a number of sophisticated segmentation algorithms relie on global thresholds as initial input parameter, such as the watershed algorithm. The main idea of our work is to provide a practical investigation on real rocks including typical artefacts affecting the reconstructed intensity images to various degrees rather than an extensive evaluation of all existing post-processing and segmentation methods (e.g. Sezgin and Sankur, 2004a (DOI: 10.1117/1.1631315); Kaestner et al., 2008 (DOI: 10.1016/j.advwatres.2008.01.022); Iassonov et al., 2009 (DOI: 10.1029/2009WR008087); Tuller et al., 2013 (DOI: 10.2136/sssaspecpub61.c8); Chauhan et al., 2016 (10.1016/j.cageo.2015.10.013), for reviews). Our idea was to give realistic constraints on the errors involved in using global thresholding and also identifying their relation to imaging conditions. We will clarify this aspect of our work and at the same time will point out the extensive body of work on more sophisticated segmentation methods and the implications of our results for them.

5.) I am not sure if the system is capable of achieving 2.5 micrometer physical resolution. The authors should verify that by using a JIMA test pattern for the relevant micro-CT settings

A JIMA Test was performed by the manufacturer of the machine resulting in a true spatial resolution of 0.9 µm under optimal conditions. Consequently, a calculated voxel size of 2.5 µm is reliable for our system. However, the true spatial resolution at our settings and sample sizes are probably slightly lower due to parameters such as changing focal spot diameter or imperfect axis alignment. We will add this information to the manuscript.

6.) Looking at Fig. 7, from my own experience, the NLM filter should perform better.

The non-local means filter was applied with the identical settings to all intensity images. We used two different windows, with a diameter of 21 voxels for the determination of the similarity weights and a diameter of 5 voxels for the actual weighted median filtering. The applied filter operates in three dimensions with a similarity value of 0.6, which corresponds to the standard settings of Avizo Fire,

Version 9.0.1. Changing those settings will certainly influence resulting pixel values. We will add the missing information of the chosen parameters of the NLM filter.

7.) Why was the median filter applied with its impact on the sharpness on grain boundaries and the associated resolution limit issue. The best results are theoretically obtained without filtering, using a segmentation method that is robust against filtering (Berg, 2018)

For our reply please see comment 3 by reviewer 3.

I think this paper presents a careful study of the measurement of porosity in sandstones – a measurement of primary importance for the mechanical behavior and permeability. The paper is essentially what I would consider a technical note, so most of my comments are technical. The paper is interesting to publish but it has some flaws.

We thank the reviewer for this general feedback.

1.) The take away message is quite specific and quite difficult to digest. This is, in my opinion related to the lack of theoretical results regarding porosity measurement which could both render conclusions more general, but also make the paper more appealing. An example of what I'm saying is to study the effect of signal-to-noise, blurring and pore size on the error of porosity estimation on a synthetic test case with only two phases. This could easily be done by clipping correctly-calculated partial volume spheres representing pores out of a block of material (what mathematical morphologists call a meatball geometry). This will allow some expectations of how these three key parameters affect the measurement, allowing your real measurements to be put into context, and yielding useful relations about blur vs pore size for accurate porosity measurements for example. Furthermore filtering can be tested synthetically.

We absolutely agree with the reviewer on the significance of synthetic tests in the field of computer tomography. Performing an extensive set of synthetic tests, however, constitutes a research project of its own. We will thus exploit work in this direction already reported in the literature, e.g. Berg et al. 2018 (DOI . 10.1190/tle37060412.1), to an extent in the revised discussion exceeding what we have done so far.

2.) Some strange choices have been made on treatment of the x-ray tomography volumes. First of all there are some clear artefacts in the high-resolution images that are not really commented.

I'm afraid that Median and NLM are really odd choices – median filtering is really only used by professional for removing extreme values, and NLM is so non-linear that it's behaviour is difficult to characterise. It's surprising that grey-mass preserving filter such as averaging (and without thinking about it too hard Gaussian blurring should also be) would be more suitable. For more advanced filtering Bilateral filtering as discussed abundantly in the literature from the Canberra group is a much better choice since it is local.

A number of parameters are missing (structuring element for filters?).

Another key point is that a very strange choice has been made to study thresholded images. If you believe that the partial volume effect holds, and that greylevels are a linear function of voxel occupancy (reasonable) for a two-phase material you can calibrate the greylevel of pores and solids and assign a porosity to each voxel with an interpolation. This calibration will remove the thresholding step and should return much less noisy images. Again this can be proved on a synthetic example as suggested above.

We will address and discuss the clearly visible artefacts, such as motion blur, in greater detail in the revised manuscript.

For choice and parameter details of the median filter, please see our reply to comment 3 of reviewer 3.

The NLM filter represents a standard technique widely used due to its adaptive characteristics and effectiveness (e.g. Andrä et al., 2013; Andrew et al., 2014; Lai et al., 2015; Sell et al., 2016; Berg et al., 2018; Pak et al., 2019), which justifies an examination in our study. We will implement a test case for the bilateral filter in our study, because we agree with the reviewer on the importance of examining a spatially varying, edge preserving filter.

We will add the missing settings, such as kernel size and shape of the structuring elements.

Please see our reply to comment 4) of reviewer 1 regarding our focus on global thresholding methods.

When global thresholding methods fail, i.e. their uncertainty becomes large, local segmentation, subpixel and interpolation methods may improve results for specific cases at the cost of computing time, complex workflows and the potential introduction of biases e.g. by the assumption of certain characteristics for neighboring pixels or about the gradient. We will explain this aspect in greater detail in the discussion and refer to the corresponding literature, e.g. Chauhan et al., 2016 (DOI: 10.1016/j.cageo.2015.10.013) or Koornev et al., 2018 (10.1002/2018WR022886).

3.) A number of laboratory measurements have been made. This is really good to compare to the x-ray measurements, however the different measurements do not seems to be consistent between each other (different non-tomography techniques do not fall into the error band of each other). This key point needs to be analysed and discussed further... different ways of measuring the same thing must yield values within error, otherwise the same thing is not being measured (or errors are underestimated).

The reviewer is thanked for noting these subtle details of our data set. As shown in Figure 4 and Table 2 the experimentally determined total porosity is larger than the effective porosity from water imbibition as expected, because total and effective porosity are two different types of porosity. The effective porosity values deduced from gas pycnometer measurements tend to be larger than the effective porosity deduced from water imbibition probably due to capillary forces, i.e. an expected relation between the various data sets. Furthermore, the effective porosity deduced from gas pycnometer measurements is either consistent with the determined total porosity within error or falls slightly below it, again physically reasonable. The sole exception is sample B, with a total porosity of 19.4±0.5 % and an effective gas porosity of 21.6±1.1 %, which we will re-check.

4.) Synchrotrons do not generally produce monochromatic beams (wigglers and bending magnets don't in any case...), but generate enough flux that the beam can be monochromatised while keeping good performance.

We will correct the statement in the final manuscript.

5.) There is normally a negative sign in the exponent, unless your μ values are negative. It may be worth talking about the dependence of μ on x-ray energy, which will allow beam hardening to be introduced with more ease later.

We will add the missing negative sign in Equation 1 of the Lambert Beer law. In addition, we will introduce the term beam hardening at this point as suggested by the reviewer.

6.) Artifacts? Artefacts.

We will happily follow the editors instruction regarding the use of British or American orthography rules.

7.) "This non-uniqueness". What is non unique? I propose to say "this spatial averaging process" "...results in a blurring of all phase boundaries in a CT image."

This is not blurring, it is the manifestation of the discretisation of the measurement. I agree that boundaries of phases will be filled with partial volume voxels, but this is an expected result of voxel-volume averaging.

We will increase the comprehensibility by deleting the misleading wording "non-uniqueness".

8.) This sentence is not homogeneous?? "The spatial resolution of the tomographic image produced by conventional X-ray CT is limited by the diameter of the X-ray source, the X-ray photon energy, the absorption of the sample as well as the number and size of pixels on the detector" "However, the (isotropic) voxel size is the calculated size of a discrete cubic volume element in the reconstructed image, which can actually be lower than the actual spatial resolution of the image"

Sure, but it's a pity to cop out of measuring at least the radiograph resolution with standard techniques.

We will revise the section and stick to the term "true spatial resolution" to increase readability throughout the manuscript.

The reviewer is right that measuring the radiograph resolution would be an interesting addition, but porosity results are independent of the true spatial resolution. The true spatial resolution was measured once by the manufacturer using a JIMA stripe pattern at an acceleration voltage of 100 kV and a current of 200 μA resulting in a true spatial resolution limit of 0.9 μm for the CT Scanner. The information will be added in the Appendix for completeness.

9.) In my experience, off focal radiation is a detail compared to mechanical stability of imaging system. Maybe also with noting phase contrast effects?

We will add the mechanical stability as a source of inaccuracy.

10.) "Standard CT systems do not record voxel intensities in terms of any physical unit but rather map the data to a somewhat arbitrary range meeting either the data storage requirements (e.g. 0 to 65535 for common 16-bit integer dataformats) or, to ensure compatibility of images from different CT systems, transforming the data to so-called Hounsfield units or CT numbers." ...and later... "Due to the non-physical units of the image data"

Weeell, no, I don't really agree. Non-physical units means that they are units what make no physical sense (to me). Here the units of the reconstructed image are a complex energy-integrated attenuation coefficient which you cannot express the units for, but this does not mean they do not exist. 16-bit integers are only used as output formats, all reconstruction codes are at least float internally for numerical reasons, so "record voxel intensities" is also wrong.

Furthermore, a detector records, here we are reconstructing, computing, calculating, inverting but not recording.

We will improve the readability by replacing the falsely used wording "non-physical units" to define the dimensionless scaling of the reconstructed images. We will also correct the error concerning the 16-bit output format and double check the correct usage of technical terms.

11.) Aesthetic comment: please don't abbreviate sandstone to sst, why? to save 5 characters? It decreases readability for no reason.

We agree with the reviewer and will delete the abbreviation sst in the entire manuscript.

12.) Rabbani is an odd reference for the watershed...

We will add some more classical references for the watershed algorithm, e.g. Lantuéjoul, 1978 or Beucher and Lantuejoul, 1979.

13.) Reconstruction process is back projection? "Convolution and back filter" is really not clear.

We agree and will revise the sentence.

14.) "...mapped to a 16-bit integer range between zero, corresponding to the background, in our case air, and 65535, corresponding to the detection limits of the CT-Scanner..."

For quantitative studies (this like one is aiming to be) cutting the reconstructed values at zero (zero what?) is bad practice, since you miss the noise distribution below zero for the background. Re-reading the sentence, I'm afraid to say that it's all meaningless and should be re-written – zero is not defined, and the "detection limits of the CTScanner" are not well defined (with a lump of lead what are your detection limits?).

We will revise the misleading statement. The measured intensities are stored using a 16-bit integer range with values between zero and 65535. The given data range is used as optimally as possible by adapting the user dependent CT parameters such as acceleration voltage. On the one hand side we avoided an overshooting of the detector by X-rays, which pass the sample without interacting, while on the other side we had to measure a statistically sufficient number of X-rays, which became attenuated by the sample i.e. by the dense minerals. Usually the evaluable data occupies only a subrange of about zero to 10000 in the reconstructed intensity images.

15.) After citing so much work by Sheppard I'm sad to see that you are not going to try a Bilateral filter which is a clear and well-defined local filter compated to the complex and hard-to-characterise NLM filter.

Please refer to our reply to comment 2) of reviewer 2. We will include a test case for the bilateral filter.

16.) Details of NLM filter setting please.

Please see our reply to comment 6) of reviewer 1 for the details regarding settings of the NLM filter.

17.) GBS points counting way off, probably copy-paste error with below, PLEASE CHECK!

We will correct the error and include the determined porosity of 20.1 with a standard deviation of 3.4 for Bentheim (GBS) sandstone.

18.) Comment on the fact that all measurements of porosity in Table 2 with their associated errors rarely fall in the same margins! This is a clear indication that the errors are extremely

underestimated, or that these porosity measurements are in fact measuring different things. I'm sorry if this is a trivial point, but it's still very surprising. I would really like an answer to this point.

We used the caliper precision stated by its manufacturer and estimated the machining precision to arrive at an uncertainty for length measures of $0.10*10^{-5}$ m; the uncertainty in weight of $3*10^{-7}$ kg is provided by the scale manufacturer. The uncertainty in fluid density reflects repeat measurements with a pycnometer and potential temperature variations in the laboratory; the pycnometer volume has an uncertainty of $5*10-10$ m$^3$. These various errors do not have the same significance for the error propagation, an aspect that we will clarify. In addition to the formal error propagation we tried to constrain the "operator-error" or "methodological error" by repeating measurements as reflected by the standard deviation in Table 2. A comparison of these two sources of uncertainty suggests that the "operator error" exceeds the nominal uncertainty in case of grain density measurements by an order of magnitude, when performing pycnometer experiments according to DIN 18124. The uncertainty for the effective porosity measurements of the gas pycnometer depend mainly on the accuracy of the used pressure transducers. The uncertainties of our used pressure transducers are 0.02 MPa.

19.) "Images obtained using X-rays pre-hardened by filters have a higher signal to noise ratio than those taken without a filter..." Why such a general sentence so far in the document? This is not true in many cases, like in biological samples.

We would remove the conflicting statement.

20.)"absolute voxel intensities are displayed using a uniform gray value range"

This is a bit rich considering you've said that they are non-physical... See comment Above

We agree with the reviewer that this statement could be misunderstood and will therefore revise the wording. The visualized slices of the reconstructed images in Figure 6 are displayed using the intensity range between zero and 7500 to increase the brightness and to reveal characteristic differences.

21.) "The quality of images taken with a focal spot diameter of 1 μm and without X-ray filters (Figure 6, FS9) is similar to images recorded with a focal spot diameter of 5 μm using aluminum filters (Figure 6, FS8)."

This is nonsense, remove it. Blur and contrast are two independent axes of image quality – which may be coupled in your system, sure, but you really should not feel empowered to make this kind of general statement, it's utterly undefendable.

Thanks to the reviewer's explanation we reconsidered this hypothesis in the results and will delete it.

22.) Figure 6, there is a fair amount of geometrical inaccuracy in the reconstructions with small pixel sizes, which looks like motion blur, which should be mentioned (FS9, WS, OS, FS8), could this be a cause of skewness?

The mentioned artefacts are probably motion blur artefacts, but could also be related to a small centre-of-rotation error. These two artefacts can look very similar either leading to blurring that results in additional partial volume elements. We checked the histograms and no apparent difference in skewness, compared to the other image histograms is visible at the high resolutions. The higher resolutions seem to compensate for those introduced artefacts. We will include a description of the observable artefacts in the CT images.

23.) Non-local means example (c, g) in Figure 7 does not fill me with joy, it's a very strange result of filter, with some edges sharpened and some texture remaining in the grains. This is the

fundamental problem with NLM, that filtering is not homogeneous or isotropic, and there is little control of how and where the algorithm chooses to filter.

I've extracted the original image from the PDF, and run 20 iterations of Bilateral, which denoises nicely – if I can upload them with my review I will. The blurred edges due to motion blur seem to be interpreted in a strange way by the filter.

We certainly agree that filtering is challenging and still a matter of research. Our result for the NLM filter is related to the chosen settings, such as similarity value and kernel size. The visual impression of the effect of a filter may not be fully reflect a filter's performance regarding quantification of porosity, as we show with our results. We will explore the bilateral filter, as noted above.

24.) Figure 8 shows a very dramatic change of the automatic thresholds. In some cases the second peak is very hard to see – adding the log of the pixel count (which is quite typical) would help in this regard.

We will modify Figure 8 and add the log of the pixel count.

25.) "In general, the difference between lab and CT porosity increases with increasing sample size and decreasing porosity."

This is pretty much expected since you are using a thresholding approach.

We agree that this is an expected result, but feel that stating it introduces nicely the following paragraph. We will revise the leading sentence and emphasize the compliance with expectations.

26.) On page 15 line 10, I think a more theoretical discussion about grey-scale-preservation in filtering (not guaranteed by either of the filters used, unlike an average filter) will help make the discussion more precise.

We will expand the discussion of filtering (see also above, e.g., inclusion of the bilateral filter).

27.) "We therefore conclude that filtering generally does not improve porosity determination."

Woah there, the choice of two strange filters does not allow such a general conclusion!!

We will remove this statement that is indeed too general and will address the complex interplay of the various settings, e.g. 2D/3D, kernel size etc.

The work focuses on the effects of image resolution, segmentation and filtering methods on the evaluation of bulk porosity in different porous sandstones through CT imaging. I think the paper represents a good contribution to scientific progress, as it outlines the limitations of x-ray microtomography in evaluating bulk porosity. However, I think their data analysis and discussion could be improved substantially. In particular, the choice of the applied methods can be debated: there are many more robust segmentation algorithms and filtering methods that would perform better and will therefore have a different impact on the final porosity quantification. The choice of the authors for these methods have not been explained, hence their final conclusions are weak. Overall, the presentation of the data is well structured. Some figures could be improved (see specific comments). Some sentences can be structured better – they are hard to read for a non-expert in the XCT field (see specific comments).

We thank the reviewer for the general judgement.

1.) First at all, the true spatial resolution (which is different from the voxel size) may depend on the sample type (e.g. grain size, composition), sample size, X-ray spot size and the precision of the alignment of the source, rotation axis and camera as well as the geometric magnification given by the ratio of distances between the source, sample and camera. Check Feser et al. (2008). In Line 9 -16, I think you should clarify better the difference between voxel size and true spatial resolution. Diameter of the x-ray source, do you mean the x-ray spot size? Please be consistent in terminology.

Thank you for pointing out the potential misunderstanding between voxel size and spatial resolution. To clarify the difference between voxel size and true spatial resolution we will explain the two terms in greater detail and add additional references, such as Feser et al (2008). We will simplify the terminology as suggested by the reviewer and use only the terms true spatial resolution and voxel size throughout the manuscript.

2.) The choice of the segmentation methods is a bit strange. If the aim of the paper was to assess the effects of different segmentation methods, you cannot base your findings only on global thresholding methods.

Line 5 should be tied up better to previous comment. In other words, why did you choose only global thresholding algorithms? Your choice is not explained

Line 34 – 35: Segmentation is the process to identify and separate phases, but it is not just based on threshold values. There are many algorithms (Indicator Kriging, C-means clustering, machine-learning algorithms such as the Trainable Weka, or watershed) that takes into account other parameters, in addition to the intensity. This misleads the reader, as the text here implies that segmentation is only possible though global thresholding, which is not true. In fact, paragraph in Line 5 should be tied up better to previous comment. In other words, why did you choose only global thresholding algorithms? Your choice is not explained.

Please see also our reply to comment 4) of reviewer 1 regarding choice of segmentation methods and aim of our study. We will clarify our focus on global thresholding methods in the revised manuscript.

Thanks for pointing out the difference between segmentation and thresholding and the misleading implication that segmentation is only possible through global thresholding. We will avoid such confusion by revising the statement.

3.) I also disagree on the choice of the filtering methods. First at all, the median filter heavily smooths the edges of the objects, and the image becomes less sharp, impacting in the resolution. In addition, a kernel size of 3 is quite large – what does it mean that you applied the filter in 3 different successions? Is that referring to 3 different kernel sizes? And what about the impact of different kernel size or different neighbouring (6 vs 26)?

The authors should consider that also these parameters may have an impact in the final quantification (and they usually do).

In Figure 7, I would expect the NLM filter to perform better – what about the influence of the choice of parameters for the filter? Did the authors verify this?

There are many other denoising filters, that are definitely a better choice than the median filter, such as bilateral filter or Anisotropic Diffusion. In addition, the results of the filtering will significantly depend also on the choice of the parameters i.e. number of iterations.

We decided to implement the median filter because of its low computing-power requirements and well known behavior.

For choice and setting details of the NLM filter, please refer to reviewer 1 comment 6. Please also refer to our reply to comment 2) of reviewer 2.

Furthermore, we (as noted above) we will include a test case for the bilateral filter in our study, because we agree on the relevance of examining a spatially varying, edge preserving filter.

4.) Something minor to consider, but since you extrapolated subvolumes, would pores lying on the edges have an effect on the final quantifications? It would be interesting to repeat the analyses, by using "exclude edges" or "border kill" in Avizo.

Also, I would add error bars in any quantification: since you are trying to assess the effect of different segmentation methods, when applying these methods your choice of the threshold is also subjected to error, therefore error bars are needed in order to estimate these errors. By using morphological operators such as erosion and dilation. See Fusseis et al (2012) for example.

Page 27, line 3 "We therefore conclude that filtering generally does not improve porosity determination" – Well, as mentioned in the points above, such a statement is odd considering that only 2 filters (and a mix of the 2) were chosen, while some more robust denoising filters are out there and should be included in this analysis for a more coherent discussion.

We would expect no difference for the determination of porosity due to pores lying at the edges of the subvolumes, because we applied only global thresholding methods using the intensity histogram. However, boundary effects will certainly play an important role when performing morphological measurements such as grain size distribution or subsequent modelling such as permeability based on the segmented pore space features.

Using morphological features is a good idea to show the uncertainty of applied thresholding methods, which we will implement for completeness.

We will revise this very general statement to the applied filters and will point out the complex interplay of the various settings, e.g. 2D/3D, kernel size etc.

**Specific comments:**

1.) Avoid sentences like "as one would" or "as one distinguishes" – always better to use a third person narrative style.

Thank you very much for pointing this out. We follow the suggestion of the reviewer and will change such inappropriate language usage.

2.) Line 15: "Owing to its relevance"?

In our opinion "owing to its relevance" is a synonym for "because of its relevance" and can be used in such context. We are not quite sure what the reviewer is aiming at.

3.) Page 3 line 3: reconstruction artifacts? I think it is artefacts.

We agree with the reviewer and will delete the word "reconstruction" before "artefacts" on Page 3 Line 2.

4.) You should explain what a voxel is, when introducing this term for the first time. A reader non-expert in the field would not know what a voxel is.

We will explain the term "voxel" when it occurs for the first time. We will modify the sentence so, hopefully, a non-expert would also understand.

5.) Page 4 line 18: what is sst? It took me a while to understand what it means, and please avoid using acronyms that i) are not scientific ii) you did not explain earlier on.

We agree and will replace the usage of the abbreviation by sandstone in the entire manuscript.

6.) Page 4 Line 19: how did you check for this? what if anisotropies were present at the microscale?

We checked macroscopically for inhomogeneity such as variations in grain size, presence of fractures and veins, layering etc. We also analyzed the SEM images. A few SEM images contained micro-fractures, which were macroscopically not visible. Thus anisotropies present at microscale cannot be excluded and could be expected for some rocks, e.g. Ruhr sandstone. These features can result in porosity variations of subvolumes. We will emphasize that we tried to address this point by the relatively large number of subvolumes investigated for every sample.

7.) Is 6.6 mm2 a representative area?

The investigated area is a compromise between sufficient resolution to determine the pores by point counting and investigating a significant fraction of the sample. For a mean grain diameter of 200 µm this area corresponds to approximately 50 grains. To account for uncertainty and to statistically validate the measurements we performed the analysis ten times on every thin section at different locations resulting in covering at least 500 grains. Furthermore, we counted at least 1000 points for each rock sample, which gives typical standard deviations between 0.5 and 4 % for our dataset depending on rock type. We find that standard deviations are larger for samples with higher porosities. According to the simple reliability chart for judging point counting results of Plas,1965 (DOI: 10.2475/ajs.263.1.87) this corresponds to a 95 % confidence interval.

8.) Assuming the samples are isotropic: what if they are not? In other words, it should have been tested.

First, the determination of the volume property, porosity, is not affected by anisotropy. The selected 9 subvolumes in the reconstructed datasets constrain the variability of the pore space. In fact, only minor differences in porosity were found between the various subvolumes, except for algorithms SH and MP, (see Figure 8). The isotropy of samples is also confirmed by our SEM analysis. A representation of heterogeneity is certainly not a problem for images of Fontainebleau sandstone (FS), but probably difficult for Ruhr sandstone (RS), due to the resolution limit.

We will add references to the isotropy of the rocks used in our study.

Generally, anisotropy may lead to attenuation variations along different directions creating additional noise and artefacts during scanning and reconstruction. Subsequent post-processing methods such as bilateral filtering or the NLM filter can be affected and results can differ as a function of the degree of anisotropy. We will address this complication in the revised manuscript.

9.) Page 7 line 11-13: how did you account for errors derived from these measurements? Or are they extremely precise? Not an expert on the matter.

The measurement errors of all our laboratory measurements were assessed by error propagation errors (Page 8, Line 14-15). All details are stated in Appendix 2. Please see also our reply to comment 18) of reviewer 2.

> 10.) Page 8 line 21: "Each pixel in the detector has an edge length of 127 μm and a gray-value resolution of 14 bit resulting in a nominal system resolution of around 1 μm" This sentence originally confused me – is it really necessary? I think it would be better to keep either nominal resolution or true spatial resolution, but not both as the reader may get confused.

We will revise the section and stick to the term "true spatial resolution" to avoid confusion.

> 11.) Page 8 line 26: How is the resolution controlled by the magnification? Get equation (see Feser et al. 2008). Also, why is this mentioned here and not at the beginning, with the rest? Be consistent.

The corresponding formula will be shifted to the introduction following the reviewer's suggestion. The section will be modified and re-structured to clarify the difference between true spatial resolution and voxel size.

> 12.) Page 9 line 20: I would put in brackets the real volume dimensions either in mm3 or um3.

We divided the reconstructed datasets into nine cuboid subvolumes with a size of 350x400x500 voxels corresponding to volume dimensions of 1mm$^3$, 35 mm$^3$ and 745 mm$^3$. We will add the missing information.

> 13.) Page 13, lines 21-25: this sentence is an interpretation and should be part of the discussions.

The interpretation will be shifted to the discussion according to the reviewer's suggestion.

> 14.) Page 14 I forgot what B, GBS and PS indicate. I would probably put the label next to them. Also, later in line 10 – you first describe the sandstones by using their relative content in porosity (highly porous sandstones) but then you call Group C, and I found this confusing: either you use the group names (since you just defined them) or the porosity content, just be consistent and do not mix between one definition and the other when listing them.

We will add the labels in brackets next to the sample name and refer to the categorized groups introduced.

> 15.) Line 13: the mean diameter falls below the resolution – does that mean that there is a lot of porosity missing from the CT analyses? And 211 designated pores – are these derived from SEM? As I am getting confused.

Sorry for the confusion we created. When the mean diameter is below the resolution limit a high fraction of porosity will be missed. We used the SEM images for pore-size analysis as an independent technique with a higher resolution than the tomographic device. We will clarify this aspect in the revision.

> 16.) Figure 4, error bars: have these been defined in the text? I did not find it, maybe I missed it.

Once again our apologies for this omission. We will clarify the meaning of error bars for each figure in the its caption.

> 17.) Figure 5: The eq. diameter here is calculated from SEM images, so it is a 2D calculation. At the same time, you mentioned the voxel size of the XCT and make a comparison between the voxel size and the mean pore diameter. However, the mean pore diameter is a 2D

measurement and therefore this comparison in my opinion does not make much sense: it would be better to use the eq diameter derived from XCT images or make a comparison of the eq diameters derived from SEM and XCT. In that sense, porosity is not the only parameter to determine, but also pore throats, pore connectivity and permeability

We use the SEM images as an independent measure for space characertization, because of its higher resolution compared to X-ray tomographic images.

The SEM images represent 2D images and consequently calculations are only in 2D. The resulting void-size distributions provide a basis for the assessment of feature size important for the evaluation of porosity from the tomographic images and are not intended for a precise geometrical characterization of the pore space in 3D.

Determining an equivalent diameter from the X-ray tomographic images limited in resolution and based on assumptions for differentiating between pores and pore throats would therefore not substantially enhance the understanding of uncertainty related to total porosity estimates from X-ray CT. Performing a comparison between 2D and 3D pore space morphology is certainly a nice suggestion for another new project, which we will keep in mind.

18.) Table 2: P-wave velocity – existing measurements from where?? Or did you calculate them?

We performed the p-wave measurements by ourselves using piezo transducers pneumatically pressed to the sample ends. Arrival times of transmitted signals were picked with a precision of 0.02 µs for P-waves and 0.06 µs for S-waves. Velocities were calculated by the ratio between sample length and travel time accounting for the intrinsic delay time of the setup. Only samples with a diameter of 10 and 30 mm were measured in the two-sensor setup to avoid damage to the fragile cores with 3 mm diameter. Uncertainty of the ultrasonic velocities were constrained by Gaussian error propagation.

19.) Figure 6: I think it should include the type of filter used, otherwise the reader has to go back and forth to table B1. A figure should be self-contained.

The reconstructed images in Figure 6 show the unfiltered intensity images. We will add the missing information in the figure caption. Furthermore, we will state the metal plates used for pre hardening the X-rays.

20.) Figure 8: Where is GBS? In line 27 page 20 you refer to GBS in figure 8 I think.

We admit that referencing of Bentheim sandstone (GBS) as an example for group A rocks in this context was misleading, because Figure 8 shows Pfaelzer sandstone (PS). These two rocks have similar porosities and exhibit comparable bimodal intensity distributions. We will include the introduced group labels in Figure 8 for reference. Furthermore, we will revise the paragraph on Page 20 by referring to the group labels instead to single rocks to improve readability and clarity throughout the manuscript.

21.) Page 28 Lines 12 – 19: For very heterogeneous samples, scientists normally adopt a multi-scale imaging approach to determine realistic values of porosity and permeability, overcoming the issue of low resolution or not representative samples. Something to keep in mind for the discussion perhaps.

Thanks for this valuable hint. We will add some references to the discussion on multi-scale imaging techniques" e.g. Okabe and Blunt, 2007 (DOI: 10.1029/2006WR005680); Ghous et al., 2008 (DOI: 10.2523/IPTC-12767-MS); Bultreys et al., 2015 (10.1016/j.advwatres.2015.05.012); Desbois et al., 2016 (DOI: 10.1016/j.petrol.2016.01.019)."

---

## Author Comment (AC3) · 4 May 2019

Please find our answers to the comments and the revised manuscript in the Supplement.

Please also note the supplement to this comment:
https://www.solid-earth-discuss.net/se-2019-48/se-2019-48-AC3-supplement.pdf